# Decomposing Policy Optimization into Proxy Objective and Knowledge Distillation for Long-context Reasoning

## Abstract

Reinforcement learning (RL) paradigms have demonstrated remarkable success in enhancing the complex reasoning capabilities of Large Language Models (LLMs). However, models trained exclusively on short-context tasks often struggle to generalize their capabilities to longer documents. Extending RL to long-context settings remains a fundamental challenge, primarily due to the high computational cost and deteriorating quality of the sampling reasoning trajectories. To bridge this gap, we propose **De**composed **P**olicy **O**ptimization (**DePO**), a novel framework that decouples policy optimization for long-context reasoning into two parallel parts: (1) leveraging short-context environments as a tractable proxy objective to acquire high-quality and efficient reasoning patterns via RL; and (2) transferring these capabilities to long-context settings through knowledge distillation, enabling the model to replicate its refined short-context reasoning strategies over extended contexts. Empirical evaluations across multiple long-context document question-answering benchmarks show that DePO consistently outperforms all baseline approaches. Notably, compared to naive long-context reinforcement learning, DePO achieves 2.8% improvement in performance while also reducing training time overhead by 49.5%. Furthermore, DePO demonstrates strong generalization capabilities, compatibility with various RL algorithms, and consistent improvements across multiple base models, offering a scalable and efficient solution for advancing long-context reasoning.

## 1 Introduction

Long-context reasoning remains a core challenge for Large Language Models (LLMs) (Kuratov et al., 2024; Ling et al., 2025; Ding et al., 2025). Conventional approaches often rely on synthesizing long-context reasoning datasets and fine-tuning models thereon (Li et al., 2024; Chen et al., 2025a; Sun et al., 2025; Chen et al., 2025b; Yang et al., 2025). However, these methods typically rely on static datasets and optimization objectives, which fail to adequately simulate or optimize the complex, dynamic decision-making processes essential for long-term reasoning. Inspired by recent successes of Reinforcement Learning (RL) in enhancing advanced reasoning capabilities in various complex tasks (DeepSeek-AI Team, 2025; OpenAI Team, 2024; Qwen Team, 2025; Chu et al., 2025), the research paradigm for long-context reasoning is increasingly shifting toward RL-based methods. Notably, emerging research suggests that this approach effectively improves model performance, demonstrating considerable potential (Wan et al., 2025; Kimi Team, 2025).

Current mainstream RL approaches, such as PPO (Schulman et al., 2017) and its variants (Yu et al., 2025; Zheng et al., 2025; Shrivastava et al., 2025) generally follow a pipeline wherein trajectories are sampled from the current policy to construct empirical data, which is subsequently used for policy updates. As a result, this paradigm introduces two critical dependencies: first, the quality of these sampled reasoning trajectories is paramount for effective policy improvement (He et al., 2025); second, the efficiency of the sampling process directly dictates the overall training overhead (Kong et al., 2025). However, both dependencies perform poorly in long-context scenarios. As illustrated in Fig. 1, compared to short-context settings, we observe that: (1) model responses exhibit substantial degradation (Fig. 1a and 1b), and (2) the computational expense of sampling increases considerably (Fig. 1c). These issues collectively lead to the suboptimal performance and

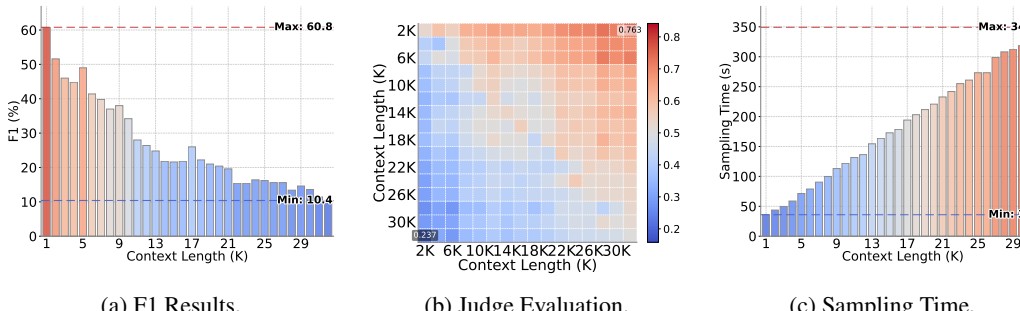

(a) F1 Results.          (b) Judge Evaluation.          (c) Sampling Time.

Figure 1: The effect of context length on Qwen2.5-7B-Instruct. The evaluation uses 500 samples from MusiQue (Trivedi et al., 2022) with synthetic contexts generated via RULER (Hsieh et al., 2024). Subfig. 1a plots F1 score versus context length. Subfig. 1b shows the confusion matrix for pairwise win rates in linguistic quality. Each cell $(i, j)$ shows the win rate of model at context length $j$ over that at length $i$. (See Appendix C.4). Subfig. 1c shows average sampling latency in seconds.

substantial computational costs of reinforcement learning in long-context environments. In contrast, such problems are much less pronounced in short-context scenarios.

Building on prior discussion, we claim that the challenging long-context optimization can be effectively reframed as a straightforward composite process: short-context optimization coupled with short-to-long distillation. Our core insight is **transfer the robust reasoning capabilities acquired in short-context optimization to enhance long-context scenarios**. Correspondingly, we propose a novel training framework, termed **De**composed **P**olicy **O**ptimization (**DePO**). Specifically, DePO comprises two parallel processes: (1) we perform RL optimization exclusively in short-context environments as a proxy objective, enabling the model to learn robust local reasoning without the confounding factor of context length; (2) simultaneously, we leverage high-quality trajectories derived from short-context optimization to transfer such acquired reasoning capabilities to long-context scenarios via knowledge distillation. Consequently, our approach encourages the final policy to achieve performance on long-context scenarios that approximates the performance of a policy trained under short-context when evaluated on this condition. Intuitively, this objective is superior to direct optimization in long-context settings, since policies optimized on short-context tasks generally achieve higher rewards on their original tasks.

To validate the effectiveness of our proposed method, we conducted a rigorous evaluation on a comprehensive long-context benchmark encompassing both moderate-context (1K–3K) and extended-context (5K–30K) scenarios. Experimental results demonstrate that DePO achieves an average performance improvement of 7.7% compared to methods using synthetic data for supervised fine-tuning or preference optimization. Furthermore, it outperforms reinforcement learning methods directly applied to long-context scenarios by 2.8% while reducing time overhead by 49.5%. This indicates that, through its unique two-stage collaborative design, DePO not only significantly enhances the model's core performance in long-context scenarios but also effectively improves training efficiency. Moreover, DePO exhibits strong generalization capabilities, seamlessly integrating with various reinforcement learning algorithms, while demonstrating stable and consistent performance across different base models. Overall, our DePO provides an efficient, scalable, and practical solution for extending the capability of LLMs to process long-context in the future.

## 2 RELATED WORK

**Long-Context Alignment.** Due to practical demands, numerous recent works have emerged that aim to enable large models to handle long-context (Dong et al., 2024; Peng et al., 2024; Xiao et al., 2024; Xiong et al., 2024a). However, maintaining reliability and faithfulness over long contexts remains a fundamental challenge (Liu et al., 2024). Consequently, long-context alignment has emerged as a critical research area dedicated to mitigating these deficiencies. Existing methodologies can be broadly categorized into two primary paradigms: data-centric and objective-focused approaches. The data-centric paradigm leverages advanced LLMs to synthesize high-

quality instruction-following data, which is then used with SFT or preference optimization for alignment (Chen et al., 2024; Bai et al., 2024a; Li et al., 2024; Chen et al., 2025b; Tang et al., 2025; Zhang et al., 2025; Zhu et al., 2025; Yang et al., 2025). In contrast, objective-focused methods aim to design more effective learning objectives, such as novel loss functions (Wu et al., 2024; Wang et al., 2025), regularization terms (Du et al., 2025), or preference alignment mechanisms (Chen et al., 2025a). Within this paradigm, the concept of short-to-long alignment has recently garnered significant attention (Du et al., 2025). For instance, LongPO (Chen et al., 2025a) demonstrates that preference data derived from short-context can be effectively leveraged to optimize long-context model via preference optimization, enabling self-improvement through an internal transfer of capabilities. Similarly, SoLoPO (Sun et al., 2025) explicitly models the relationship between short and long inputs by decomposing long-context preference optimization into learning preference signals from short contexts and performing short-to-long reward alignment. However, the short-to-long transfer for online RL is largely underexplored, and our work fills this gap.

**Reinforcement Learning.** RL paradigm has emerged as a powerful paradigm for advancing the complex reasoning abilities of LLMs. This line of research, pioneered by foundational methods like GRPO (Shao et al., 2024; DeepSeek-AI Team, 2025), has inspired a suite of subsequent refinements, including DAPO (Yu et al., 2025), GPG (Chu et al., 2025) and GSPO (Zheng et al., 2025). Despite these successes, the application of such techniques remains largely confined to short-context scenarios, such as mathematical reasoning tasks. How to effectively extend RL to long-context reasoning remains a significant challenge. Although some studies have attempted to adapt RL to long-context settings and have demonstrated preliminary potential (Huang et al., 2025; Li et al., 2025; Wan et al., 2025), these approaches often overlook the inherent difficulties of long-context environments, resulting in limited performance gains and high computational costs. In contrast, our work integrates a short-context reinforcement learning objective acting as a proxy, combined with knowledge distillation to transfer reasoning capabilities, offering an efficient and effective solution.

## 3 METHODOLOGY

In this section, we first provide an overview of several prominent reinforcement learning algorithms that serve as the foundation for our method (§3.1). Building on this, we introduce our novel training paradigm, Decomposed Policy Optimization, which we detail in the subsequent section (§3.2).

### 3.1 PRELIMINARIES

**Group Relative Policy Optimization (GRPO).** GRPO (Shao et al., 2024) is a reinforcement learning algorithm specifically developed to improve the reasoning abilities of LLMs. It optimizes the policy by directly computing a relative advantage score across a set of candidate responses to the same query, thereby eliminating the need for a value model. Specifically, for a given question $x$, it begin by sampling a set of $G$ candidate responses $\{o_i\}_{i=1}^{G}$ from an old policy $\pi_{\theta_{\text{old}}}$. Each response $o_i$ is subsequently evaluated by a reward function to obtain a corresponding reward $r_i$. The optimization objective for the policy $\pi_\theta$ is then formulated as:

$$\mathcal{J}_{\text{GRPO}}(\theta) = \mathbb{E}_{x \sim \mathcal{D}, \{o_i\}_{i=1}^{G} \sim \pi_{\theta_{\text{old}}}(\cdot|x)} \left[ \frac{1}{G} \sum_{i=1}^{G} \frac{1}{|o_i|} \sum_{t=1}^{|o_i|} \left( \min\left( w_{i,t}(\theta) A_{i,t}, \right. \right. \right.$$
$$\left. \left. \left. \text{clip}\left( w_{i,t}(\theta), 1 - \varepsilon, 1 + \varepsilon \right) A_{i,t} \right) - \beta \mathbb{D}_{\text{KL}}(\pi_\theta || \pi_{\text{ref}}) \right) \right], \tag{1}$$

where $\pi_\theta$ is the current policy model, $\pi_{\theta_{\text{old}}}$ is the old policy model before updating, $\varepsilon$ is the clipping hyperparameter, the importance ratio $w_{i,t}(\theta)$ and the advantage $A_{i,t}$ for token $o_{i,t}$ are given by:

$$w_{i,t}(\theta) = \frac{\pi_\theta(o_{i,t} \mid x, o_{i,<t})}{\pi_{\theta_{\text{old}}}(o_{i,t} \mid x, o_{i,<t})}, \quad A_{i,t} = \frac{r_i - \text{mean}(\{r_k\}_{k=1}^{G})}{\text{std}(\{r_k\}_{k=1}^{G})}. \tag{2}$$

DAPO (Yu et al., 2025) enhances the stability and efficiency of reinforcement learning for LLMs through several key contributions. First, it employs a higher clipping threshold to mitigate policy entropy degradation. Second, it improves training efficiency via a dynamic sampling method that

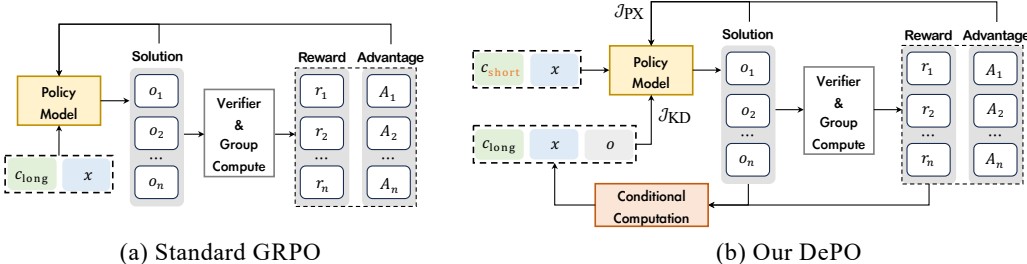

(a) Standard GRPO             (b) Our DePO

Figure 2: Comparison of our DePO method with Standard GRPO in long-context optimization.

filters out instances with zero reward variance. Finally, to address sequence length bias, DAPO introduces a token-level loss function to penalize overly short outputs, complemented by a reward shaping strategy that discourages excessively long generations.

**Group Sequence Policy Optimization (GSPO).** The key innovation of GSPO (Zheng et al., 2025) lies in its theoretically grounded formulation of the importance ratio based on sequence-level likelihoods, which aligns with the fundamental principle of importance sampling. Furthermore, GSPO employs normalized rewards derived from the advantages across multiple responses to a given query, thereby ensuring consistency between sequence-level reward assignment and policy optimization. GSPO optimizes the following sequence-level objective:

$$\mathcal{J}_{\text{GSPO}}(\theta) = \mathbb{E}_{x \sim D, \{o_i\}_{i=1}^G \sim \pi_{\theta_{\text{old}}}(\cdot|x)} \left[ \frac{1}{G} \sum_{i=1}^{G} \min \left( s_i(\theta)\hat{A}_i, \text{clip}(s_i(\theta), 1-\varepsilon, 1+\varepsilon)\hat{A}_i \right) \right], \quad (3)$$

where the importance ratio $s_i(\theta)$ is defined using sequence likelihood:

$$s_i(\theta) = \left( \frac{\pi_\theta(o_i|x)}{\pi_{\theta_{\text{old}}}(o_i|x)} \right)^{\frac{1}{|o_i|}} = \exp \left( \frac{1}{|o_i|} \sum_{t=1}^{|o_i|} \log \frac{\pi_\theta(o_{i,t}|x, o_{i,<t})}{\pi_{\theta_{\text{old}}}(o_{i,t}|x, o_{i,<t})} \right). \quad (4)$$

### 3.2 DECOMPOSED POLICY OPTIMIZATION FOR LONG-CONTEXT REASONING

#### 3.2.1 OVERALL FRAMEWORK

In Fig. 2, we provide a high-level illustration of our approach and contrasts it with standard GRPO. Our core insight lies in grounding reinforcement learning within simplified scenarios to ensure efficient reward acquisition, and then extending these capabilities via knowledge distillation. Building on this, we introduce DePO that decomposes the complex end-to-end optimization of long-context reasoning into two more tractable objectives.

First, RL optimization is constrained to short-context scenarios, thereby developing a proxy policy proficient in localized reasoning. Subsequently, the reasoning expertise acquired from short contexts is transferred to long contexts via knowledge distillation, enabling the policy to effectively extrapolate its abilities to extended sequences. Notably, these two phases can be seamlessly integrated into existing RL frameworks through the composite objective formulation presented below,

$$\mathcal{J}(\theta) = \mathcal{J}_{\text{PX}}(\theta) + \lambda \mathcal{J}_{\text{KD}}(\theta), \quad (5)$$

where $\lambda$ is a hyperparameter that modulates the weight between the two objective terms. This combined loss ensures that the policy not only improves its performance via RL in shorter episodes but also aligns its behavior with the distilled knowledge to maintain consistency and generalization across longer contexts.

#### 3.2.2 REINFORCING SHORT-CONTEXT REASONING AS PROXY OBJECTIVE

In our DePO framework, we train policy by restricting the reinforcement learning process solely to short-context scenarios. The underlying intuition is that by reducing the complexity of the problem space, the model can more effectively acquire fundamental reasoning capability, without being influenced by the confounding factor of context length.

---

**Algorithm 1:** Decomposed Policy Optimization (DePO)

---

**Input:** Initial policy parameters $\theta_{\text{init}}$, dataset $\mathcal{D}$, learning rate $\eta$, group size $G$, distillation weight $\lambda$, batch size $B$

**Output:** Optimized policy parameters $\theta$

1 Initialize policy parameters by $\theta \leftarrow \theta_{\text{init}}$

2 **for** *each training step* **do**

    `// Sample a mini-batch of data`

3     Sample a batch $\mathcal{B} = \{(c_{\text{short}}^{(j)}, c_{\text{long}}^{(j)}, x^{(j)}, y^{(j)})\}_{j=1}^{B}$ from $\mathcal{D}$

4     Initialize batch losses $\mathcal{L}_{\text{PX}} \leftarrow 0$, $\mathcal{L}_{\text{KD}} \leftarrow 0$

    `// Part 1: Reinforcing Short-context Reasoning as Proxy Objective`

5     **for** *each data point $j \in \{1, \ldots, B\}$ in parallel* **do**

        `// Generate rollouts using Short Context`

6         Generate $G$ rollouts $\{o_{j,i}\}_{i=1}^{G}$ using $\pi_\theta(\cdot \mid x^{(j)}, c_{\text{short}}^{(j)})$

7         Compute rewards $\{r_{j,i}\}_{i=1}^{G}$ where $r_{j,i} = \text{EM}(o_{j,i}, y^{(j)})$

8         Compute advantages $\{A_{j,i}\}_{i=1}^{G}$ (normalized within group $j$)

9     **end**

    `// Calculate proxy policy loss`

10    Compute the proxy loss $\mathcal{L}_{\text{PX}}(\theta)$ using Eq. 1 or Eq. 3

    `// Part 2: Empower Long-context Reasoning via Knowledge Distillation`

    `// Filter high-quality trajectories across the entire batch`

11    Create a set of teacher trajectories $\widehat{\mathcal{O}}_{\text{batch}} = \{o_{j,i} \mid r_{j,i} = 1, \forall j \in \mathcal{B}, \forall i \in \{1..G\}\}$

    `// Calculate knowledge distillation loss`

12    Compute $\mathcal{L}_{\text{KD}}(\theta)$ based on $\widehat{\mathcal{O}}_{\text{batch}}$ using Eq. 10

    `// Combine losses and update policy once per batch`

13    Compute compound loss: $\mathcal{L}(\theta) = \mathcal{L}_{\text{PX}}(\theta) + \lambda \mathcal{L}_{\text{KD}}(\theta)$

14    Update policy parameters: $\theta \leftarrow \theta - \eta \nabla_\theta \mathcal{L}(\theta)$

15 **end**

---

Specifically, each data point consists of a tuple $(c_{\text{short}}, c_{\text{long}}, x) \in \mathcal{D}$, where $c_{\text{short}}$ refers to a shortened version of the supporting documents for question $x$, and $c_{\text{long}}$ denotes a longer context with additional information. The process starts with an old policy $\pi_{\theta_{\text{old}}}$ generating a set of $G$ (i.e., the group size) rollouts $\{o_i\}_{i=1}^{G}$ conditioned on a short context:

$$o_i \sim \pi_{\theta_{\text{old}}}(\cdot \mid x, c_{\text{short}}). \tag{6}$$

The reinforcement learning proxy objective $\mathcal{J}_{\text{PX}}(\theta)$ is then optimized using these short-context rollouts. When combined with GRPO, it can be formally expressed as:

$$\mathcal{J}_{\text{PX}}(\theta) = \mathbb{E}_{(x, c_{\text{short}}) \sim \mathcal{D}, \{o_i\}_{i=1}^{G} \sim \pi_{\theta_{\text{old}}}(\cdot | x, c_{\text{short}})} \left[ \frac{1}{G} \sum_{i=1}^{G} \frac{1}{|o_i|} \sum_{t=1}^{|o_i|} \left( \min \left( w_{i,t}(\theta) \hat{A}_{i,t}, \right. \right. \right.$$
$$\left. \left. \left. \text{clip}\left(w_{i,t}(\theta), 1 - \epsilon, 1 + \epsilon\right) \hat{A}_{i,t} \right) \right) \right], \tag{7}$$

where the importance ratio $w_{i,t}(\theta)$ and the advantage $A_{i,t}$ for token $y_{i,t}$ are given by:

$$w_{i,t}(\theta) = \frac{\pi_\theta(o_{i,t} \mid x, c_{\text{short}}, o_{i,<t})}{\pi_{\theta_{\text{old}}}(o_{i,t} \mid x, c_{\text{short}}, o_{i,<t})}, \quad A_{i,t} = \frac{r_i - \text{mean}(\{r_k\}_{k=1}^{G})}{\text{std}(\{r_k\}_{k=1}^{G})}. \tag{8}$$

**Reward Design.** During the training process, the reward function plays a critical role in guiding the policy model's behavior. We employ the *Exact Match (EM)* metric as the reward signal. This metric assigns a reward of 1 if the generated response exactly matches the reference answer, and 0 otherwise, providing an unambiguous learning signal that encourages accuracy in the model's outputs.

### 3.2.3 EMPOWER LONG-CONTEXT REASONING VIA KNOWLEDGE DISTILLATION

Traditional knowledge distillation transfers capabilities from a powerful teacher model to a compact student. In our framework, we adapt this technique to align the model's behavior under long-context

Table 1: Detailed statistics of our train and test datasets. Length is calculated by the Qwen tokenizer.

| Statistics | Train | | Moderate-context | | | Extended-context | | | | | | |
|---|---|---|---|---|---|---|---|---|---|---|---|---|
| | Short | Long | 2Wiki | HQA | Musi | DocMath | Frames | 2Wiki | HQA | Musi | NarQA | Qasp |
| # Examples | 5000 | 5000 | 12,576 | 7,405 | 2,417 | 200 | 824 | 200 | 200 | 200 | 200 | 200 |
| Avg. Length | 1,218 | 14,814 | 1,004 | 1,417 | 2,524 | 17,570 | 15,712 | 7,491 | 13,391 | 16,287 | 29,848 | 5,035 |
| Max. Length | 1,529 | 14,957 | 4,352 | 3,687 | 5,304 | 176,004 | 117,078 | 16,995 | 17,599 | 17,842 | 65,319 | 21,888 |

settings with its proficiency under short-context conditions. Specifically, we further leverage high-quality responses generated from a short-context optimization process to function as supervisory signals to transfer knowledge to the model in long-context scenarios.

To achieve this, we minimize the forward Kullback–Leibler (KL) divergence to align the response distributions. The forward KL divergence quantifies the information loss when approximating a target distribution $P$ with a candidate distribution $Q$. It is defined as:

$$\mathbb{D}_{\mathrm{KL}}(P \parallel Q) = \mathbb{E}_{x \sim P}\left[\log \frac{P}{Q}\right]. \tag{9}$$

Minimizing this divergence encourages student to assign high probability to actions deemed favorable by the teacher. In our DePO, the teacher is defined as the policy conditioned on the short context, denoted as $P = \pi_\theta(\cdot \mid c_{\mathrm{short}}, x)$, while the student model refers to the same policy conditioned on the long context, expressed as $Q = \pi_\theta(\cdot \mid c_{\mathrm{long}}, x)$. Crucially, the rollouts generated during the short-context optimization can be directly leveraged for knowledge distillation at each step. This naturally motivates the forward KL divergence objective:

$$\mathcal{J}_{\mathrm{KD}}(\theta) = -\mathbb{D}_{\mathrm{KL}}(\pi_\theta(\cdot|c_{\mathrm{short}}, x) \| \pi_\theta(\cdot|c_{\mathrm{long}}, x)),$$

$$= \mathbb{E}_{(c_{\mathrm{short}}, c_{\mathrm{long}}, x) \sim \mathcal{D}, \{o_i\}_{i=1}^G \sim \pi_\theta(\cdot|c_{\mathrm{short}}, x)}\left[\log \frac{\pi_\theta(o_i|c_{\mathrm{long}}, x)}{\pi_\theta(o_i|c_{\mathrm{short}}, x)}\right]. \tag{10}$$

To provide a theoretical justification for this design, we introduce the following lemma:

**Lemma 3.1** (Policy Generalization Gap). *Let $\mathcal{J}_{short}(\theta)$ and $\mathcal{J}_{long}(\theta)$ denote the expected returns of policy $\pi_\theta$ under short-context and long-context conditions, respectively. For any policy $\pi_\theta$, the absolute difference in expected returns between the two context settings is bounded by the square root of the expected KL divergence between the corresponding conditional output distributions:*

$$|\mathcal{J}_{short}(\theta) - \mathcal{J}_{long}(\theta)| \le \sqrt{\frac{1}{2} \cdot \mathbb{E}_{(x, c_{short}, c_{long}) \sim \mathcal{D}}\left[D_{KL}\left(\pi_\theta(\cdot \mid x, c_{short}) \| \pi_\theta(\cdot \mid x, c_{long})\right)\right]}. \tag{11}$$

The proof of Lemma 3.1 is provided in Appendix A.1. This result indicates that minimizing the KL divergence between the policy's output distributions under short and long contexts reduces the performance gap between the two settings, thereby offering a theoretical justification for the proposed distillation approach. Additionally, we present further discussion in Appendix A.2 on the advantages of DePO compared to directly applying reinforcement learning to long-context models.

**Conditional Computation.** Since reward values are computed for all rollouts during proxy training, we further incorporate conditional computation to focus on sequences that yield high rewards or are labeled as correct. This selective approach enhances efficiency and stability of knowledge transfer by prioritizing a high-quality teacher trajectory. The conditional objective can be expressed as:

$$\mathcal{J}_{\mathrm{CondKD}}(\theta) = \mathbb{E}\left[\log \frac{\pi_\theta(y_i \mid c_{\mathrm{long}}, x)}{\pi_\theta(y_i \mid c_{\mathrm{short}}, x)} \;\middle|\; r(y_i) \ge \tau\right]. \tag{12}$$

In our approach, we simply set $\tau = 1$, corresponding to the case of an exact match.

## 4 EXPERIMENTS

### 4.1 EXPERIMENTAL SETUP

**Dataset Construction.** To generate training data with controlled context lengths, we followed the procedure outlined in RULER (Hsieh et al., 2024). Using the supporting documents and question-

Table 2: Main results across comprehensive long-context DocQA benchmarks. All models are fine-tuned based on the Qwen2.5-7B-Instruct. **Bold** values indicate the best performance, while underlined ones denote the second-best results. † denotes DePO was trained using DAPO.

| Methods | Moderate-context | | | Extended-context | | | | | | | |
|---|---|---|---|---|---|---|---|---|---|---|---|
| | 2Wiki | HQA | Musi | DocMath | Frames | 2Wiki | HQA | Musi | NarQA | Qasp | Avg. |
| Base Model | 47.8 | 65.2 | 36.1 | 27.0 | 24.4 | 45.8 | 52.8 | 25.0 | 17.6 | 34.1 | 37.6 |
| *Superivised Fine-tuning* | | | | | | | | | | | |
| vanilla SFT | 48.9 | 67.1 | 40.1 | 15.0 | 30.0 | 45.1 | 61.0 | 31.1 | 26.5 | 37.9 | 40.3 |
| LongMIT | 47.3 | 62.0 | 35.4 | 36.5 | 36.6 | 43.3 | 54.6 | 32.6 | 26.0 | 44.9 | 41.9 |
| LongReward | 48.0 | 64.6 | 36.1 | 28.5 | 37.8 | 43.4 | 54.5 | 32.7 | **27.8** | 44.3 | 41.8 |
| SeaLong-SFT | 47.2 | 64.6 | 35.6 | 36.5 | 36.6 | 41.6 | 51.3 | 29.8 | 25.9 | 42.1 | 41.1 |
| LongFaith-SFT | 57.7 | 63.7 | 42.7 | 25.5 | 28.8 | 45.2 | 44.1 | 28.4 | 23.1 | 43.7 | 40.3 |
| Pos2Distill-$R^2$ | 71.4 | 72.8 | 55.5 | 40.5 | 28.7 | 66.0 | 48.5 | 36.0 | 12.0 | 37.8 | 46.9 |
| *Preference Optimization* | | | | | | | | | | | |
| LongPO | 47.8 | 64.2 | 31.5 | 22.0 | 22.9 | 42.3 | 54.1 | 17.8 | 18.0 | 31.4 | 35.2 |
| SoLoPO | - | - | - | - | - | 63.3 | 58.8 | **50.7** | 26.1 | 43.0 | - |
| SeaLong-PO | 46.6 | 64.0 | 36.0 | 39.0 | 36.9 | 41.0 | 52.4 | 32.3 | 26.0 | **45.1** | 41.9 |
| LongFaith-PO | 48.6 | 65.7 | 37.8 | 20.5 | 23.5 | 42.8 | 50.2 | 25.3 | 17.4 | 34.1 | 36.6 |
| *Reinforcement Learning* | | | | | | | | | | | |
| RAG-RL | 70.8 | 68.9 | 47.2 | - | - | - | - | - | - | - | - |
| GRPO | 66.3 | 65.8 | 42.3 | **43.0** | 39.4 | 59.0 | 53.4 | 38.6 | 24.2 | 42.0 | 47.4 |
| DAPO | 71.8 | 72.7 | 53.5 | 42.0 | **42.6** | 66.7 | 58.7 | 45.9 | 24.4 | 39.5 | 51.8 |
| GSPO | 69.8 | 72.8 | 52.4 | 40.0 | 39.1 | 64.5 | **62.0** | 44.2 | 27.1 | 41.4 | 51.3 |
| DePO† | **74.4** | **74.1** | **59.1** | **43.0** | 41.8 | **73.4** | 60.2 | 49.5 | 26.8 | 43.4 | **54.6** |

answer pairs from the MuSiQue (Trivedi et al., 2022) training dataset as our foundation, we constructed both short and long contexts for each sample. Specifically, the short context environment $c_{short}$ was generated by augmenting the set of essential golden supporting documents with a controlled number of randomly sampled irrelevant distractors, after which the document order was randomized. To construct the long context environment $c_{long}$, we treated the short context as a baseline superset, appending additional distractor documents, followed by a similar randomization of the document sequence. This synthesis process yielded a final dataset $\mathcal{D}$ comprising 5,000 training samples, each formalized as a tuple $(c_{short}, c_{long}, x, y)$, with detailed statistics provided in Table 1. The resulting short contexts $c_{short}$ average ~1K, while the long contexts $c_{long}$ average ~15K.

**Comparison Methods.** To systematically evaluate the effectiveness of our proposed DePO method on long-context reasoning tasks, we conduct a comprehensive comparative analysis against a series of advanced approaches spanning three mainstream methodologies: Supervised Fine-tuning, Preference Optimization, and Reinforcement Learning. Under Supervised Fine-tuning, we compare with vanilla SFT, SeaLong-SFT (Li et al., 2024), LongMIT (Chen et al., 2025b), LongReward-SFT (Zhang et al., 2025), LongFaith-SFT (Yang et al., 2025), and Pos2Distill-$R^2$ (Wang et al., 2025). For Preference Optimization, the compared methods include LongPO (Chen et al., 2025a), SoLoPO (Sun et al., 2025), SeaLong-PO, and LongFaith-PO. Under Reinforcement Learning, we consider RAG-RL (Huang et al., 2025), GRPO (Shao et al., 2024), DAPO (Yu et al., 2025), and GSPO (Zheng et al., 2025), the latter three of which employ long-context training. Additional details regarding these baselines are provided in the Appendix.

**Evaluation Benchmarks.** Our evaluation is conducted on a series of long-context document question answering (DocQA) benchmarks, which we categorize into two groups based on context length. For the moderate context setting, models are evaluated on contexts averaging 1K–3K tokens, using the following benchmarks: 2WikiMultihopQA (Ho et al., 2020), HotpotQA (Yang et al., 2018), and MuSiQue (Trivedi et al., 2022). These evaluation sets are derived from the original datasets. For the extended context setting, which involves contexts with an average length ranging from 5K to 30K tokens, we include the following benchmarks: 2WikiMultihopQA, HotpotQA, MuSiQue, NarrativeQA (Kočiský et al., 2018), Qasper (Dasigi et al., 2021), Frames (Krishna et al., 2025), and DocMath (Zhao et al., 2024). Among these, 2WikiMultihopQA, HotpotQA, MuSiQue, NarrativeQA, and Qasper are sourced from LongBench (Bai et al., 2024b), Docmath is from Wan et al. (2025), and Frames is obtained from its original dataset. We report the F1-score for all benchmarks except DocMath, for which we use Accuracy. Detailed statistics are provided in Table 1.

Table 3: Performance of DePO integrated with DAPO and GSPO. All models are fine-tuned based on the Qwen2.5-7B-Instruct. † denotes DePO was trained using DAPO and ‡ denotes DePO was trained using GSPO, respectively. **Bold** values indicate top results, while underlined ones denote the second-best results. Baseline details are in Appendix C.1.

| Methods | Moderate-context | | | Extended-context | | | | | | | |
|---|---|---|---|---|---|---|---|---|---|---|---|
| | 2Wiki | HQA | Musi | DocMath | Frames | 2Wiki | HQA | Musi | NarQA | Qasp | Avg. |
| Base Model | 47.8 | 65.2 | 36.1 | 27.0 | 24.4 | 45.8 | 52.8 | 25.0 | 17.6 | 34.1 | 37.6 |
| DAPO (Short) | 66.2 | 72.2 | 57.9 | 40.5 | 39.9 | 56.1 | 59.8 | 45.4 | 23.5 | 31.8 | 49.3 |
| DAPO (Long) | 71.8 | 72.7 | 53.5 | 42.0 | 42.6 | 66.7 | 58.7 | 45.9 | 24.4 | 39.5 | 51.8 |
| DAPO (Short-Long) | 65.2 | 70.8 | 52.7 | 43.0 | 43.1 | 62.1 | 60.3 | 49.2 | 26.0 | 38.9 | 51.1 |
| DePO† | **74.4** | 74.1 | **59.1** | 43.0 | 41.8 | **73.4** | 60.2 | 49.5 | **26.8** | 43.4 | 54.6 |
| DePO-IW† | 74.1 | **74.7** | 57.1 | **44.0** | **43.3** | 71.8 | **63.6** | 51.6 | 26.4 | 43.8 | **55.0** |
| GSPO (Short) | 63.1 | 68.2 | 53.3 | 39.5 | 39.2 | 63.0 | 61.3 | 42.7 | 25.2 | 39.6 | 49.5 |
| GSPO (Long) | 69.8 | 72.8 | 52.4 | 40.5 | 39.1 | 64.5 | **62.0** | 44.2 | 27.1 | 41.4 | 51.4 |
| GSPO (Short-Long) | 70.5 | 71.6 | 53.1 | **43.5** | 41.4 | 61.5 | 57.0 | 44.7 | 23.2 | 39.3 | 50.6 |
| DePO‡ | 73.0 | 73.7 | **55.4** | 41.0 | 42.3 | 72.4 | 59.6 | 49.2 | **27.4** | 43.8 | 53.8 |
| DePO-IW‡ | **73.7** | **74.3** | 55.1 | 42.5 | **43.8** | **73.7** | 61.6 | 49.0 | 26.1 | 43.5 | **54.3** |

## 4.2 MAIN RESULTS

**SOTA performance on long-context DocQA with Qwen2.5-7B-Instruct.** We conducted an extensive evaluation of various long-context methodologies across both moderate and extended context scenarios. Our empirical findings indicate that DePO achieves the highest average performance across all evaluated models, demonstrating a substantial advantage over all other methods. Specifically, DePO outperforms SFT-based methods by 7.7%, surpasses

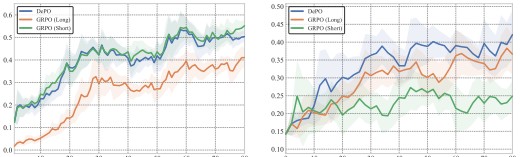

(a) Training Rewards.    (b) Validation Accuracy.

Figure 3: Training dynamics of DePO with baselines under DAPO.

PO methods by 12.7%, and exceeds RL-based methods by 2.8%. Furthermore, we visualized the training process, as illustrated in Fig. 3. Empirical observations indicate that reinforcement learning applied to long-context tasks typically results in slower reward convergence. In contrast, our method exhibits a reward progression during training that is consistent with RL in a short-context setting. This suggests that the inherent simplicity of short-context tasks facilitates more stable and efficient convergence compared to long-context training. Notably, evaluation accuracy on the long-context validation set demonstrates that our approach significantly enhances capacity in long-context, surpassing methods that apply RL directly to long-contexts.

**DePO delivers consistent performance improvements when integrated with diverse reinforcement learning algorithms and foundation models.** As evidenced in Tables 3 and 4, DePO achieves average gains of +2.0%, +2.8%, and +2.4% when combined with GRPO, DAPO, and GSPO, respectively. These results demonstrate the versatility of DePO, which reliably enhances core capabilities in long-context reasoning without dependence on any specific RL algorithm. Furthermore, we applied our method to multiple foundation models. As shown in Table 4, DePO yields absolute improvements of +2.0%, +2.6%, +5.0%, and +2.4% in average score on Qwen2.5-7B-Instruct, Qwen2.5-3B-Instruct, Llama3.2-3B-Instruct (Llama Team, 2024), and Llama3.1-8B-Instruct, respectively. These consistent gains across models of varying scales provide strong evidence for the generalizability of our DePO approach.

**Importance-weighted off-policy reinforcement learning further enhances DePO.** Alternatively, the distillation process can be formulated as an off-policy reinforcement learning objective. Specifically, the high-quality trajectories $\{o_i\}_{i=1}^{G}$ generated by the policy under short contexts are regarded as off-policy data (Yan et al., 2025). We then employ importance sampling to estimate the expected value under the policy $\pi_\theta$ when conditioned on the corresponding long context $c_{\text{long}}$, using trajectories generated under the short context $c_{\text{short}}$. This approach enables policy updates aimed at improving long-context reasoning without requiring

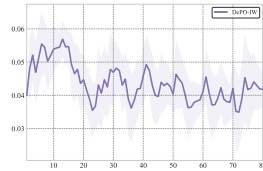

Figure 4: PG Clip Fraction in off-Policy RL.

Table 4: Performance of DePO integrated with various base models. DePO was trained using GRPO. **Bold** values indicate the best performance, while underlined ones denote the second-best results.

| Methods | Moderate-context | | | Extended-context | | | | | | | |
|---|---|---|---|---|---|---|---|---|---|---|---|
| | 2Wiki | HQA | Musi | DocMath | Frames | 2Wiki | HQA | Musi | NarQA | Qasp | Avg. |
| Qwen2.5-7B-Instruct | 47.8 | 65.2 | 36.1 | 27.0 | 24.4 | 45.8 | 52.8 | 25.0 | 17.6 | 34.1 | 37.6 |
| + vanilla SFT | 48.9 | 67.1 | 40.1 | 25.5 | 30.0 | 45.1 | **61.0** | 31.1 | **26.5** | 37.9 | 41.3 |
| + GRPO (Short) | 64.2 | 68.1 | 53.4 | 37.0 | 34.9 | 59.5 | 55.6 | 42.6 | 19.6 | 24.6 | 46.0 |
| + GRPO (Long) | 66.3 | 65.8 | 42.3 | **43.0** | 39.4 | 59.0 | 53.4 | 38.6 | 24.2 | 42.0 | 47.4 |
| + GRPO (Short-Long) | 71.9 | **74.2** | 51.8 | 39.5 | 39.0 | 66.4 | 58.4 | 42.3 | 25.1 | 38.0 | 50.7 |
| + DePO | **72.1** | 73.1 | **55.7** | 41.0 | 42.5 | 66.4 | 57.8 | 48.5 | 26.4 | 43.1 | **52.7** |
| Qwen2.5-3B-Instruct | 41.0 | 55.3 | 25.4 | 21.5 | 21.3 | 32.9 | 38.8 | 18.1 | 15.4 | 29.3 | 29.9 |
| + vanilla SFT | 39.6 | 54.4 | 29.8 | 16.5 | 18.8 | 34.2 | 39.8 | 19.8 | 4.6 | 29.5 | 28.7 |
| + GRPO (Short) | 47.8 | 58.6 | 41.1 | 14.5 | 10.4 | 25.3 | 39.5 | 32.2 | 20.5 | 35.6 | 32.6 |
| + GRPO (Long) | 49.3 | 59.2 | 40.3 | 27.0 | 32.2 | 45.9 | 41.5 | 34.4 | 22.5 | 34.9 | 38.7 |
| + GRPO (Short-Long) | 49.0 | **62.4** | 42.5 | 23.5 | 15.5 | 41.2 | 44.7 | 36.1 | 20.9 | 17.2 | 35.3 |
| + DePO | **52.0** | 61.3 | 42.5 | 29.5 | 32.9 | 47.3 | 46.3 | 40.4 | 22.8 | 38.0 | 41.3 |
| Llama3.2-3B-Instruct | 41.7 | 61.5 | 29.5 | 24.0 | 31.2 | 34.6 | 48.4 | 28.5 | 14.5 | 28.2 | 34.2 |
| + vanilla SFT | 42.5 | 63.5 | 39.8 | 22.0 | 30.9 | 37.6 | 51.5 | 28.9 | 15.0 | 36.3 | 36.8 |
| + GRPO (Short) | 55.6 | 65.8 | 47.0 | 22.0 | 30.9 | 39.1 | 42.4 | 24.8 | 18.6 | 35.5 | 38.2 |
| + GRPO (Long) | 51.4 | 63.6 | 48.7 | 23.0 | 39.9 | 53.0 | 53.6 | 44.1 | 20.6 | 40.4 | 43.8 |
| + GRPO (Short-Long) | 50.6 | 64.5 | 47.4 | 23.5 | 35.9 | 49.3 | 48.5 | 40.8 | 16.8 | 25.0 | 40.2 |
| + DePO | **65.8** | **70.0** | **56.9** | 32.0 | 39.2 | 64.4 | 53.4 | 42.9 | 23.1 | 40.0 | 48.8 |
| Llama3.1-8B-Instruct | 53.6 | 69.5 | 43.1 | 33.0 | 34.7 | 49.7 | 54.6 | 33.4 | 25.4 | 34.1 | 43.1 |
| + vanilla SFT | 59.7 | **73.5** | 53.1 | 14.0 | 39.8 | 54.6 | **64.7** | 40.8 | **33.2** | 40.7 | 47.4 |
| + GRPO (Short) | 68.8 | 72.6 | 53.4 | 38.5 | 37.4 | 66.1 | 54.2 | 39.5 | 9.3 | 30.5 | 47.0 |
| + GRPO (Long) | 70.2 | 73.1 | 57.1 | 40.0 | 46.5 | 60.7 | 58.4 | 47.8 | 25.8 | 25.9 | 50.6 |
| + GRPO (Short-Long) | 69.1 | 71.1 | 52.9 | 41.0 | 39.7 | 60.5 | 46.6 | 35.2 | 25.7 | 23.4 | 46.5 |
| + DePO | **70.4** | **73.5** | 53.1 | 41.0 | 46.2 | 67.3 | 56.9 | 48.1 | 28.9 | 44.1 | **53.0** |

Table 5: Evaluation results of DePO on the QAs-RULER and LongBench V2 benchmarks.

| Methods | QAs-RULER | | | | | LongBench V2 | | | | | |
|---|---|---|---|---|---|---|---|---|---|---|---|
| | 4K | 8K | 16K | 32K | Avg. | Easy | Hard | <32K | 32K-128K | >128K | Overall |
| Qwen2.5-7B-Instruct | 65.5 | 63.4 | 51.5 | 30.9 | 52.8 | 30.9 | 28.3 | 36.9 | 24.6 | 26.1 | 29.3 |
| + DePO | 68.8 | 63.8 | 57.8 | 55.9 | 61.6 | 37.5 | 31.5 | 38.9 | 28.8 | 35.2 | 33.8 |
| Llama3.1-8B-Instruct | 61.6 | 56.4 | 36.8 | 14.6 | 42.4 | 31.8 | 28.6 | 38.3 | 25.1 | 25.0 | 29.8 |
| + DePO | 68.0 | 62.3 | 63.2 | 58.9 | 63.1 | 34.9 | 29.3 | 37.8 | 26.0 | 31.5 | 31.4 |

additional online interaction. We reformulate the importance ratio from Eq. 1 and 3 as follows:

$$\widetilde{w}_{i,t}(\theta) = \frac{\pi_\theta(o_{i,t} \mid x, c_{\text{long}}, o_{i,<t})}{\pi_{\theta_{\text{old}}}(o_{i,t} \mid x, c_{\text{short}}, o_{i,<t})}, s_i(\theta) = \exp\left(\frac{1}{|o_i|} \sum_{t=1}^{|o_i|} \log \frac{\pi_\theta(o_{i,t}|x, c_{\text{long}}, o_{i,<t})}{\pi_{\theta_{\text{old}}}(o_{i,t}|x, c_{\text{short}}, o_{i,<t})}\right). \quad (13)$$

We refer to this variant as DePO-IW. The complete algorithm of DePO-IW can be found in the Appendix 2. As shown in Table 3, incorporating offline importance sampling yields further performance gains. When integrated into the DAPO, DePO-IW surpasses the baseline by 3.2% and the original DePO by 0.4%. This advantage is consistent when DePO-IW is applied to the GSPO algorithm, yielding improvements of 2.9% over the baseline and 0.5% over DePO. We further illustrate the training dynamics of DePO-IW. As shown in Fig. 4, the policy gradient clipping ratio during the offline reinforcement learning phase remains consistently low throughout training, generally below 5.0%. This ensures the effectiveness of the training process for DePO-IW (Fu et al., 2025).

## 4.3 QUANTITATIVE ANALYSIS

**Impact of context length and task difficulty.** To systematically evaluate our proposed DePO on question-answering tasks involving documents of varying lengths and difficulty levels, we conduct experiments on the RULER and LongBench V2 (Bai et al., 2025) with YaRN (Peng et al., 2024) benchmarks. As shown in Table 5, experimental results indicate that as context length increases, the performance of base models on RULER declines significantly, with a pronounced drop observed at the 32K length. After integrating DePO, both models consistently improve across all context lengths, effectively mitigating the performance degradation under long-context settings (16K and 32K). On the more challenging LongBench V2 benchmark, DePO also demonstrates strong generalization

capabilities. For hard tasks, the model achieves stable performance gains, with the most substantial improvements observed in contexts exceeding 128K. For example, Qwen2.5-7B improves from 26.1% to 35.2%. These results indicate that DePO not only enhances the model's long-context processing capabilities but also significantly boosts its reasoning ability, leading to improved overall performance in real-world, complex long-document QA scenarios.

**Effect of distillation context length.** To investigate the impact of distillation context length on model performance and training efficiency, we fix the short context length at 1K and conducted experiments under long context settings of 4K, 8K, 12K, and 15K tokens. Our approach is compared against baseline models trained with GRPO at corresponding context lengths. As illustrated in Fig. 5, our method consistently and significantly outperform the baselines across all tested context lengths while substantially reducing training time. For instance, at a context

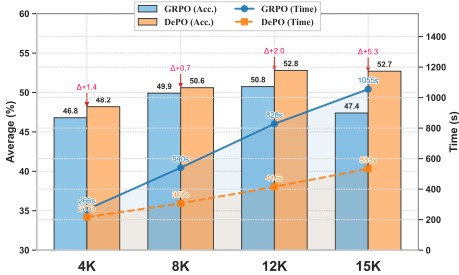

Figure 5: Impact of training context length.

length of 15K, our approach achieves a 5.3% improvement in performance alongside a 49.5% reduction in training time compared to GRPO. Furthermore, we observe a trade-off in the baseline GRPO method: model performance improves continuously as context length increases from 4K to 12K, but declines when further extended to 15K, likely due to degradation in response quality caused by excessively long contexts. In contrast, our method demonstrates consistent superiority across all context lengths, confirming its effectiveness and robustness in long-context scenarios.

**Evaluation of DePO on general short-context tasks.** Prior studies have suggested that enhancing the long-context capabilities of language models may degrade their performance on short-context tasks (Xiong et al., 2024b; Dong et al., 2025). To assess the effectiveness of our method in short-context scenarios, we conduct experiments on a suite of benchmarks (see Appendix C.3 for details). As shown in Fig. 6, the results demonstrate that DePO achieves these extensions without compromising short-context performance. Notably, DePO not only preserves general knowledge but also yields av-

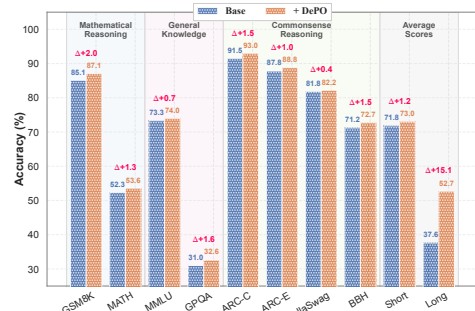

Figure 6: Short-context Evaluation.

erage improvements of 1.6% on mathematical reasoning and 1.1% on commonsense reasoning tasks.

## 5 CONCLUSION

In this work, we propose Decomposed Policy Optimization, a novel framework that decouples the end-to-end long-context optimization process into proxy objective and knowledge distillation. Our empirical evaluations across a comprehensive suite of long-context document question-answering benchmarks demonstrate the superiority of DePO. This method consistently outperforms established baselines, while also achieving a significant reduction in training overhead. In summary, our work provides a practical and effective pathway for developing more powerful and reliable long-context reasoning models. Furthermore, we believe this decomposed optimization paradigm holds significant promise beyond its current application, offering a versatile strategy for tackling complex problems in other domains.

## REPRODUCIBILITY STATEMENT

We are committed to ensuring the transparency and reproducibility of our work. To facilitate independent verification, we provide a comprehensive description of our experimental pipeline in Sec. 4.1 and Appendix C. This includes details on dataset preparation, model configuration, training procedures, as well as the algorithmic workflow and a complete list of hyperparameters and implementation specifics.

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

# A  THEORETICAL ASNALYSIS

## A.1  PROOF OF POLICY GENERALIZATION GAP

> **Lemma** (Policy Generalization Gap). *Let $\mathcal{J}_{short}(\theta)$ and $\mathcal{J}_{long}(\theta)$ denote the expected returns of policy $\pi_\theta$ under short-context and long-context conditions, respectively. For any policy $\pi_\theta$, the absolute difference in expected returns between the two context settings is bounded by the square root of the expected KL divergence between the corresponding conditional output distributions:*
>
> $$|\mathcal{J}_{short}(\theta) - \mathcal{J}_{long}(\theta)| \le \sqrt{\frac{1}{2} \cdot \mathbb{E}_{x \sim \mathcal{D}} \left[ D_{KL} \left( \pi_\theta(\cdot|x, c_{short}) \, \| \, \pi_\theta(\cdot|x, c_{long}) \right) \right]}. \tag{14}$$

***Proof.*** The proof proceeds by first relating the difference in expected returns to the Total Variation (TV) distance between the policy's output distributions, and then bounding the TV distance using the Kullback-Leibler (KL) divergence via Pinsker's inequality.

First, we formally define the expected returns. The expected return $\mathcal{J}(\theta)$ is the expectation of the reward $R(x, o)$ over the distribution of inputs $(c, x) \sim \mathcal{D}$ and the policy's output distribution $o \sim \pi_\theta(\cdot|x, c)$.

For the short-context and long-context settings, the expected returns are:

$$\mathcal{J}_{\text{short}}(\theta) = \mathbb{E}_{(c_{\text{short}}, x) \sim \mathcal{D}, \, o \sim \pi_\theta(\cdot|x, c_{\text{short}})}[R(x, o)] \tag{15}$$

$$\mathcal{J}_{\text{long}}(\theta) = \mathbb{E}_{(c_{\text{long}}, x) \sim \mathcal{D}, \, o \sim \pi_\theta(\cdot|x, c_{\text{long}})}[R(x, o)] \tag{16}$$

For notational simplicity, let's denote the conditional policies as $\pi_{\text{short}}(o|x) = \pi_\theta(o|x, c_{\text{short}})$ and $\pi_{\text{long}}(o|x) = \pi_\theta(o|x, c_{\text{long}})$. The absolute difference in expected returns can be written as:

$$|\mathcal{J}_{\text{short}}(\theta) - \mathcal{J}_{\text{long}}(\theta)| = \left| \mathbb{E}_{x \sim \mathcal{D}} \left[ \mathbb{E}_{o \sim \pi_{\text{short}}(\cdot|x)}[R(x, o)] \right] - \mathbb{E}_{x \sim \mathcal{D}} \left[ \mathbb{E}_{o \sim \pi_{\text{long}}(\cdot|x)}[R(x, o)] \right] \right|$$

$$= \left| \mathbb{E}_{x \sim \mathcal{D}} \left[ \mathbb{E}_{o \sim \pi_{\text{short}}(\cdot|x)}[R(x, o)] - \mathbb{E}_{o \sim \pi_{\text{long}}(\cdot|x)}[R(x, o)] \right] \right| \tag{17}$$

By applying Jensen's inequality ($|\mathbb{E}[X]| \le \mathbb{E}[|X|]$), we can move the absolute value inside the outer expectation:

$$|\mathcal{J}_{\text{short}}(\theta) - \mathcal{J}_{\text{long}}(\theta)| \le \mathbb{E}_{x \sim \mathcal{D}} \left[ \left| \mathbb{E}_{o \sim \pi_{\text{short}}(\cdot|x)}[R(x, o)] - \mathbb{E}_{o \sim \pi_{\text{long}}(\cdot|x)}[R(x, o)] \right| \right]. \tag{18}$$

Now, we focus on the inner term for a fixed input $x$. The absolute difference in expected rewards is bounded by the product of the reward range and the Total Variation (TV) distance between the two probability distributions. The TV distance between two distributions $P$ and $Q$ over the same space $\mathcal{A}$ is defined as $D_{\text{TV}}(P\|Q) = \frac{1}{2} \sum_{a \in \mathcal{A}} |P(a) - Q(a)|$. A standard result states that for any function $f$ with range $[m, M]$, $|\mathbb{E}_P[f] - \mathbb{E}_Q[f]| \le (M - m) D_{\text{TV}}(P\|Q)$. In our case, the function is the reward $R(x, o)$, which is bounded in $[0, 1]$. Applying this inequality, we get:

$$\left| \mathbb{E}_{o \sim \pi_{\text{short}}(\cdot|x)}[R(x, o)] - \mathbb{E}_{o \sim \pi_{\text{long}}(\cdot|x)}[R(x, o)] \right| \le D_{\text{TV}} \left( \pi_{\text{short}}(\cdot|x) \, \| \, \pi_{\text{long}}(\cdot|x) \right)$$

Substituting this result back into Eq. 18:

$$|\mathcal{J}_{\text{short}}(\theta) - \mathcal{J}_{\text{long}}(\theta)| \le \mathbb{E}_{x \sim \mathcal{D}} \left[ D_{\text{TV}} \left( \pi_{\text{short}}(\cdot|x) \, \| \, \pi_{\text{long}}(\cdot|x) \right) \right]. \tag{19}$$

Next, we relate the TV distance to the KL divergence using Pinsker's inequality, which states that for any two probability distributions $P$ and $Q$:

$$D_{\text{TV}}(P\|Q) \le \sqrt{\frac{1}{2} D_{\text{KL}}(P\|Q)}. \tag{20}$$

Applying this inequality to our conditional policy distributions for a fixed $x$:

$$D_{\text{TV}} \left( \pi_{\text{short}}(\cdot|x) \, \| \, \pi_{\text{long}}(\cdot|x) \right) \le \sqrt{\frac{1}{2} D_{\text{KL}} \left( \pi_{\text{short}}(\cdot|x) \, \| \, \pi_{\text{long}}(\cdot|x) \right)}. \tag{21}$$

Substituting this into Eq. 19:

$$|\mathcal{J}_{\text{short}}(\theta) - \mathcal{J}_{\text{long}}(\theta)| \leq \mathbb{E}_{x \sim \mathcal{D}} \left[ \sqrt{\frac{1}{2} D_{\text{KL}} \left( \pi_{\text{short}}(\cdot|x) \,\|\, \pi_{\text{long}}(\cdot|x) \right)} \right]. \tag{22}$$

Finally, we apply Jensen's inequality to the square root function:

$$|\mathcal{J}_{\text{short}}(\theta) - \mathcal{J}_{\text{long}}(\theta)| \leq \sqrt{\frac{1}{2} \cdot \mathbb{E}_{x \sim \mathcal{D}} \left[ D_{\text{KL}} \left( \pi_\theta(\cdot|x, c_{\text{short}}) \,\|\, \pi_\theta(\cdot|x, c_{\text{long}}) \right) \right]}. \tag{23}$$

This completes the proof. □

## A.2 Advantage of DePO

As detailed in Appendix A.1, we define the expected return $\mathcal{J}(\theta)$ as the expectation of the reward $R(x, o)$ over the distribution of inputs $(c, x) \sim \mathcal{D}$ and the policy's output distribution $o \sim \pi_\theta(\cdot|x, c)$.

For the short-context and long-context settings, the respective expected returns are:

$$\mathcal{J}_{\text{short}}(\theta) = \mathbb{E}_{(c_{\text{short}}, x) \sim \mathcal{D}, \, o \sim \pi_\theta(\cdot|x, c_{\text{short}})}[R(x, o)] \tag{24}$$

$$\mathcal{J}_{\text{long}}(\theta) = \mathbb{E}_{(c_{\text{long}}, x) \sim \mathcal{D}, \, o \sim \pi_\theta(\cdot|x, c_{\text{long}})}[R(x, o)] \tag{25}$$

Let $\theta_{\text{short}}$ and $\theta_{\text{long}}$ denote the parameters of policies optimized exclusively on short- and long-context data, respectively. Due to the inherent challenges of long-context modeling, it is often observed that the performance on short contexts surpasses that on long contexts:

$$\mathcal{J}_{\text{short}}(\theta_{\text{short}}) > \mathcal{J}_{\text{long}}(\theta_{\text{long}}) \tag{26}$$

The primary objective of DePO is to train a policy, parameterized by $\theta_{\text{DePO}}$, that can process long contexts with a level of performance comparable to a policy specialized for short contexts. We formalize this goal as closing the performance gap, aiming to achieve:

$$\mathcal{J}_{\text{long}}(\theta_{\text{DePO}}) \approx \mathcal{J}_{\text{short}}(\theta_{\text{short}}) \tag{27}$$

This can be expressed as:

$$\mathcal{J}_{\text{short}}(\theta_{\text{short}}) - \mathcal{J}_{\text{long}}(\theta_{\text{DePO}}) = \varepsilon, \quad \varepsilon > 0 \tag{28}$$

Here, $\varepsilon$ represents the approximation error. Crucially, this error term is directly controlled and minimized by the KD $\mathcal{J}_{\text{KD}}$ component of our optimization objective. By minimizing $\varepsilon$, we are effectively training the long-context policy towards the higher performance target set by $\mathcal{J}_{\text{short}}(\theta_{\text{short}})$. Consequently, a successfully optimized DePO policy is expected to outperform a policy trained solely on long-context data. This leads to the desired outcome:

$$\mathcal{J}_{\text{long}}(\theta_{\text{DePO}}) > \mathcal{J}_{\text{long}}(\theta_{\text{long}}) \tag{29}$$

# B Discussion on Short-to-Long Distillation

## B.1 Conceptual Framework and Core Principles

Our training paradigm synthesizes two core principles: the intrinsic invariance of reasoning patterns and the extrinsic control of the data scaffold.

### B.1.1 Invariance of Reasoning Patterns Across Context Lengths

Qualitatively, we observe that the fundamental reasoning patterns a model employs are conserved across varying context lengths. The model's reasoning trajectory can typically be decomposed into three distinct parts:

- **Sub-goal Decomposition**: Breaking down the main question into manageable sub-queries.
- **Contextual Grounding**: Locating and extracting specific evidence from the text.

- **Reasoning Summary**: Piecing together evidence to form a conclusion.

To illustrate this invariance, we compare the reasoning trajectory of **Qwen2.5-7B-Instruct** in both short and long context settings for the question: *"How many square miles in the city where KRMP is located?"*.

**Short-Context Setting**

- Sub-goal Decomposition: To determine how many square miles the city where KRMP is located covers, we need to identify the city from the given information about KRMP.

- Contextual Grounding: From the document about KRMP, it states: "KRMP (1140 AM) is an urban adult contemporary radio station in Oklahoma City, Oklahoma."

- Reasoning Summary: This sentence clearly indicates that KRMP is located in Oklahoma City.

- Sub-goal Decomposition: Next, we need to find the total area of Oklahoma City.

- Contextual Grounding: From the document about Oklahoma City, it states: "According to the United States Census Bureau, the city has a total area of 620.34 square miles (1,606.7 km2)..."

- Reasoning Summary: Therefore, the answer is 620.34 square miles.

**Long-Context Setting**

- Sub-goal Decomposition: To determine how many square miles the city where KRMP is located covers, we need to identify the city associated with KRMP from the provided information.

- Contextual Grounding: From the document, we find: "KRMP (1140 AM) is an urban adult contemporary radio station in Oklahoma City, Oklahoma."

- Reasoning Summary: This tells us that KRMP is located in Oklahoma City, Oklahoma.

- Sub-goal Decomposition: Next, we need to find the total area of Oklahoma City. From the information provided: "According to the United States Census Bureau, the city has a total area of 620.34 square miles..."

- Reasoning Summary: Therefore, the total area of the city where KRMP is located (Oklahoma City) is 620.34 square miles.

This comparison highlights that the fundamental, step-by-step reasoning process is invariant to context length. The difference in performance stems from the increased difficulty of locating evidence and the degradation of reasoning abilities in longer contexts. DePO is designed to capitalize on this invariance. By training on short contexts, we enable the model to learn these core reasoning structures in a clean environment, free from the noise of extensive documents.

## B.2 ISOLATING REASONING FROM DISTRACTION VIA CONTEXTUAL SCAFFOLDING

To facilitate the transfer of reasoning abilities acquired in short contexts to their application in long contexts, we employ a deliberate data construction strategy that serves as a form of contextual scaffolding.

- **Short-context Environment Construction.** Following RULER, we begin with the set of golden support documents essential for answering a given question. We then construct the short context by augmenting this essential set with a controlled number of irrelevant distractor documents until a moderate length limit is reached. Then, the order of all documents within this context is then randomized.
- **Long-context Environment Construction.** The long context is constructed as a superset of the short context. We take the total documents from $c_{\text{short}}$ (both support and distractor) and append additional distractor documents to reach much longer context length. The order of documents is also randomized.

Based on this, we ensure that the fundamental evidence and the corresponding reasoning structure remain invariant across different context instances.

## B.3 EMPIRICAL SUPPORT FROM TRAINING DYNAMICS

Beyond this conceptual framework, our empirical results provide strong quantitative evidence for this generalization. As shown in Fig. 4, which analyzes the training dynamics of our off-policy RL variant (DePO-IW), the policy gradient clipping fraction remains consistently low throughout training. In off-policy RL, a low clipping ratio is a strong indicator that the trajectories sampled

from the proxy policy (the policy conditioned on $c_{\text{short}}$) are highly compatible with the target policy's objective (the policy conditioned on $c_{\text{long}}$).

### B.4 ESTABLISHED PRECEDENT FOR SHORT-TO-LONG GENERALIZATION

In addition, the paradigm of short-to-long generalization is gaining significant traction, with recent works like LongPO and SoLoPO demonstrating its effectiveness in preference optimization. These methods validate the core idea that capabilities learned from shorter inputs can be transferred to improve long-context performance. However, extending this paradigm to online RL remains largely unexplored, which involves active trajectory sampling and dynamic policy updates. DePO is designed to fill this specific gap, offers a sound and effective framework.

## C IMPLEMENTATION DETAILS

In this section, we present the experimental setup used in our study, covering both the training and evaluation settings of the models.

### C.1 BASELINES

- **LongMIT (Chen et al., 2025b):** This paper presents the Multi-agent Interactive Multi-hop Generation framework, designed to construct a high-quality instruction dataset for multi-hop reasoning. The framework employs specialized agents to generate foundational single-hop question-answer pairs from a diverse corpus. These pairs are subsequently sampled based on semantic relevance and merged to form complex multi-hop queries requiring information synthesis. To ensure high fidelity, this work introduces a dedicated Quality Verification Agent to evaluate the generated content using a scoring mechanism and filters out suboptimal samples.

- **LongReward (Zhang et al., 2025):** The data construction involves a two-stage process for creating datasets for Supervised SFT and DPO. The SFT dataset is initiated by generating diverse question-answering pairs from 10,000 long-context documents via the Self-Instruct technique, using the GLM-4 pre-training corpus and model. This is subsequently augmented with 76k instances from ShareGPT. For preference optimization process, the dataset is constructed by sampling ten responses per prompt. These responses are then scored using GLM-4 as the judge, to form preference pairs by selecting the highest- and lowest-scoring responses for training.

- **SeaLong-SFT (Li et al., 2024):** SEALONG synthesizes training data via a self-supervised pipeline. The process commences with query-document pairs from the MuSiQue training set, where long contexts are constructed by augmenting relevant documents with randomly sampled irrelevant ones. For each instance, the model generates a candidate pool of multiple reasoning trajectories using temperature sampling. The core of the method lies in a self-evaluation mechanism operationalized through Minimum Bayes Risk, where each candidate is scored based on its semantic consistency with the others in the pool, as measured by sentence embedding similarity. This consensus-based scoring identifies the most plausible reasoning paths, which are then used to create supervision data. The final data can be formatted either as single high-scoring exemplars for supervised fine-tuning or as preference pairs that contrast high-scoring and low-scoring outputs for preference optimization.

- **Pos2Distill-R$^2$ (Wang et al., 2025):** The data construction process begins by identifying an advantageous position, which empirically determined to be the recent slots in the context. The base model is then prompted with the relevant documents placed in this advantageous configuration to generate high-quality reasoning trajectory. This response serves as teacher output for training. Subsequently, for the same query, multiple trivial input prompts are constructed. In these prompts, the same set of critical documents are placed at various other positions within the context window, which are known to induce performance degradation. Each of these unfavorable prompts is then paired with the high-quality reasoning trajectory generated from the advantageous position.

- **LongPO (Chen et al., 2025a):** The procedure begins by curating a long-context corpus from existing sources. For a given long document, the core of the method involves a reverse-generation process. First, a shorter self-contained chunk of text is randomly sampled from the long document. It uses self-Instruct methodology to prompt well-aligned short-context language model to generate

a relevant instruction based on this short chunk, ensuring the chunk contains all necessary information to answer the instruction. Subsequently, this model generates two responses to the instruction: a chosen response conditioned on the short relevant chunk, and a rejected response conditioned on the entire long document. This creates a preference pair that captures the discrepancy between the model's well-aligned short-context capabilities and its under-aligned long-context behavior.

- **SoLoPO (Sun et al., 2025):** The process begins with the MuSiQue multi-hop question-answering dataset. Following the context synthesis strategy from the RULER, for each question and its corresponding supporting documents, both a short context and a long context are generated. This is achieved by combining the essential supporting documents with a varying number of randomly sampled, irrelevant documents to reach the target lengths, thereby simulating the information redundancy common in real-world long texts. Subsequently, preference pairs are generated for these contexts. These generated responses are then evaluated against the answers to identify the highest-quality response as the chosen answer and a lower-quality one as the rejected answer.
- **LongFaith (Yang et al., 2025):** For the SFT dataset, the process begins by using a LLM to synthesize faithful reasoning chains. This is achieved by providing the model with a question, the answer, and the specific supporting facts from the source documents, using a chain-of-citation prompt that requires the model to provide attributions for each reasoning step. To create the preference optimization dataset, this high-faithfulness reasoning chain serves as the chosen sample, which is then contrasted with rejected samples exhibiting questionable faithfulness. These rejected samples are intentionally synthesized to model three specific failure modes, includes misinformation, lack of attribution, and potential knowledge conflicts.

## C.2  TRAINING CONFIGURATION

In our experimental setup, the model is trained using verl (Sheng et al., 2024) with a maximum sequence length of 16,384 tokens. To improve training efficiency and optimize GPU memory usage, we integrate FlashAttention 2 (Dao, 2023) and the vLLM engine (Kwon et al., 2023). Additionally, we employ Ulysses sequence parallelism (Jacobs et al., 2024) with a size set to 4. To ensure a fair comparison, for each method we select the checkpoint that achieves the highest performance on the LongBench V1 musique task within two training epoch as the final model, which is then evaluated across multiple benchmarks. All Experiments were trained on eight NVIDIA H20 96GB GPUs. Further implementation details and hyperparameter configurations can be found in Table 6, which includes the experimental setup used when integrating our method with various reinforcement learning algorithms.

Table 6: Hyperparameters for DePO and baseline methods.

| Hyper-parameter | DePO (GRPO) | DePO (DAPO) | DePO (GSPO) |
|---|---|---|---|
| Optimizer | AdamW | AdamW | AdamW |
| Learning rate | 5e-7 | 5e-7 | 5e-7 |
| KL type | K3 | - | - |
| KL coefficient | 0.001 | - | - |
| Training batch size | 64 | 64 | 64 |
| PPO mini batch size | 64 | 64 | 8 |
| Max prompt length | 15360 | 15360 | 15360 |
| Max response length | 1000 | 1000 | 1000 |
| Reward metrics | EM | EM | EM |
| Sampling temperature | 0.7 | 0.7 | 0.7 |
| Number of rollouts | 8 | 8 | 8 |
| Distillation weight $\lambda$ | 1.0 | 1.0 | 1.0 |
| Clip ratio high | 0.20 | 0.28 | 0.0004 |
| Clip ratio low | 0.20 | 0.20 | 0.0003 |

## C.3  EVALUATION CONFIGURATION

To thoroughly assess the performance of our DePO, we carry out extensive experiments on a variety of benchmark datasets, as detailed below:

### C.3.1 LONG-CONTEXT BENCHMARKS.

Given that our training data is constructed from the multihop question-answering dataset MuSiQue (Trivedi et al., 2022), with sequence lengths limited to under 16K tokens, we focus on evaluating model performance on long-context question-answering tasks within a context window of up to 32K tokens.

For the moderate-context setting, which involves documents averaging between 1K to 3K tokens, we employ the following benchmarks:

- **HotpotQA** (Yang et al., 2018) is used to evaluate the model's fundamental ability to perform two-hop reasoning and to provide sentence-level supporting evidence, thereby testing its capacity for explainable reasoning.
- **2WikiMultihopQA** (Ho et al., 2020) serves to test the model's robustness on more challenging reasoning paths where the connection between facts is implicit, thus probing its ability for more sophisticated information synthesis.
- **MuSiQue** (Trivedi et al., 2022) is employed to test the model's capacity for handling long-form, compositional inference, challenging it to navigate structured reasoning chains of up to four hops.

For the extended-context setting, which involves documents averaging between 5K and 30K tokens, we employ the following benchmarks:

- **Longbench V1** (Bai et al., 2024b) serves as a comprehensive benchmark suite designed to holistically evaluate a model's long-context proficiency. It encompasses a diverse set of tasks, such as question answering, summarization, and code completion, thereby facilitating a thorough assessment of the model's generalizability and performance across multiple domains.
- **DocMath** (Zhao et al., 2024) is used to evaluate the numerical reasoning capabilities of LLMs within realistic, expert-level scenarios. Moving beyond standard exam-like problems, this benchmark grounds complex quantitative questions in long, specialized documents that contain both unstructured text and structured tables. It is therefore designed to assess a model's ability to comprehend lengthy contexts, synthesize disparate information, and perform the multi-step reasoning required to solve challenging problems representative of real-world expert domains.
- **Frames** (Krishna et al., 2025) is used to evaluate model performance in end-to-end Retrieval-Augmented Generation (RAG) systems. It utilizes challenging multi-hop questions to assess a model's ability to retrieve, reason over, and synthesize information from multiple sources, thereby testing its capacity to generate factually grounded and coherent answers.

Additionally, we conduct qualitative analyses on two supplementary evaluation benchmarks:

- **QAs-RULER** (Hsieh et al., 2024) is employed to qualitatively assess the model's fine-grained comprehension and retrieval abilities in realistic long-context scenarios. This benchmark features "needle-in-a-haystack" tasks where the target information is subtly phrased and embedded within extensive distractors, thereby probing the model's robustness against informational noise and its capacity for semantic understanding beyond simple keyword matching.
- **Longbench V2** (Bai et al., 2025) is utilized to stress-test the model's performance at the frontiers of long-context processing. Building upon its predecessor, it introduces tasks with significantly increased context lengths and complexity, enabling a nuanced examination of how model capabilities degrade or adapt under extreme contextual pressures.

### C.3.2 SHORT-CONTEXT BENCHMARKS.

In the short-context evaluation, we assess model across comprehensive benchmarks as follows:

- **GSM8K** (Cobbe et al., 2021) assesses multi-step mathematical reasoning through a test set of 1,319 problems. Each problem requires parsing natural language questions and performing a series of arithmetic operations.
- **MATH** (Hendrycks et al., 2021b) evaluates advanced mathematical problem-solving across algebra, geometry, number theory, and precalculus. Its test set includes 5,000 problems that demand sophisticated symbolic reasoning and abstract thinking.

- **MMLU** (Hendrycks et al., 2021a) measures broad knowledge and reasoning ability through 15,900 multiple-choice questions spanning 57 subjects. Its extensive scope makes it a standard for evaluating a model's knowledge and its generalization capabilities.
- **GPQA** (Rein et al., 2024) is a challenging expert-level benchmark with 448 multiple-choice questions in biology, physics, and chemistry. It requires deep domain-specific reasoning beyond superficial knowledge retrieval.
- **ARC** (Clark et al., 2018) is a multiple-choice question-answering dataset derived from science exams for grades 3 through 9. It is partitioned into an easy set and a challenge set, with the latter comprising more difficult questions that necessitate reasoning.
- **HellaSwag** (Zellers et al., 2019) is a new challenge dataset designed to evaluate commonsense natural language inference. It is constructed via adversarial filtering, wherein a sequence of discriminators is iteratively employed to curate machine-generated incorrect answers.
- **BBH** (Suzgun et al., 2023) comprises 23 challenging tasks. Its primary objective is to assess the capacity of language models for multi-step reasoning, which presents a core challenge that extends beyond mere pattern recognition.

These benchmarks collectively evaluate mathematical reasoning, general knowledge, and commonsense reasoning capabilities. We use accuracy as the evaluation metric for all tasks, employing evaluation scripts based on OpenCompass (Contributors, 2023).

### C.4   DETAILS OF JUDGE EVALUATION

To evaluate the quality of model responses across varying context lengths, we employ Qwen-Plus as an automated judge model. In order to mitigate positional bias (Liu et al., 2024), which refers to the tendency in pairwise comparisons where the order of response presentation may affect the outcome, we implement a symmetric evaluation protocol. Specifically, for each pair of responses $(R_A, R_B)$ corresponding to different context lengths, we conduct two independent comparisons: one in the order $(R_A, R_B)$ and the other in the reversed order $(R_B, R_A)$. The final win rate is calculated by averaging the outcomes from these two comparative evaluations, thereby canceling out systematic errors introduced by ordering effects. The prompt template is as follows:

> **Prompt Templates for Judge Evaluation.**
>
> You are a distinguished computational linguist and an expert in text quality assessment. Your primary function is to conduct a pairwise comparison of two text responses (Response A vs. Response B), both generated by the same Large Language Model. Your objective is to adjudicate which response exhibits superior quality based exclusively on formal linguistic attributes.
> You must conduct your comparative analysis along the following three dimensions, adhering to the specified criteria:
> Grammar & Correctness: Assess the absolute correctness of spelling, punctuation, and syntactic structures. Evaluate whether word choice is precise and appropriate, and if the text fully adheres to the grammatical and expressive norms of standard written English. Identify any instances of malapropisms, incorrect collocations, or grammatical errors.
> Fluency & Naturalness: Assess the text's readability and natural flow. Scrutinize for any awkward or convoluted phrasing or redundancy.
> Structure & Logic: Evaluate the overall organizational architecture of the text. Analyze the logical chain of argumentation, narration, or exposition to ensure it is clear and internally consistent.

#### C.4.1   DECODING SETTINGS

We employ SGLang (Zheng et al., 2024) as the inference engine for all evaluation experiments. Decoding hyperparameters are configured according to the specific evaluation benchmark. For the moderate-context, extended-context, RULER, and short-context benchmarks, greedy decoding is utilized to ensure deterministic and optimal outputs. For the LongBenchV2 benchmark, we follow the settings specified in the original paper and set the sampling temperature to 0.1.

## D Impact of $\lambda$ on Performance of DePO

To evaluate the impact of the hyperparameter $\lambda$ in our composite objective function (Eq. 5), we conducted a series of experiments on the Qwen2.5-7B-Instruct model. As shown in Figure 7, the performance of DePO exhibits considerable robustness to the choice of $\lambda$, achieving its peak at $\lambda = 0.5$. We analyze that a sufficiently small $\lambda$ would cause the model to over-prioritize proxy optimization, leading to limited generalization to long contexts. Conversely, a large $\lambda$ might lead the model to excessively focus on knowledge distillation (KD) from long contexts, thereby neglecting improvement on short contexts. Despite this trade-off,

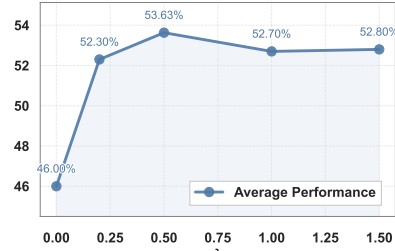

Figure 7: Performance w/ different $\lambda$.

the overall performance of DePO remains largely stable across a range of $\lambda$ values, demonstrating the method's robustness and practicality.

## E Algorithm

In this section, we present the pseudocode for DePO and its importance-weighted variant DePO-IW, detailed in Algorithm 1 and Algorithm 2, respectively. Both algorithms share a common structure involving a proxy objective based on short-context reasoning and a knowledge distillation component for long-context empowerment, yet they differ in how they utilize trajectories sampled from short-context environment for the long-context update.

## F Future work

Our training DePO objective is to enhance the model's intrinsic long-context capabilities, and this has been thoroughly validated on various long-document reasoning benchmarks. While applying our DePO directly to information-dense long-context tasks presents challenges, we believe the underlying decomposed optimization paradigm of DePO holds significant potential for a broad range of long-context related tasks:

- Repository-level code completion: The short-context could be function signatures and key dependencies, while the long-context is the entire repository.
- Object detection in computer vision: The short-context could be a cropped image containing only the target object, while the long-context is the complex original scene containing that object.

## G The use of LLM

The preparation of this manuscript was assisted by a Large Language Model (LLM) for the purpose of refining language and style, specifically to improve clarity and readability. The LLM's role was strictly limited to that of a writing enhancement tool. It made no contribution to the substantive intellectual work, which includes the conceptualization of the research, methodological design, experimental execution, data analysis, and interpretation of the results. These core components were the exclusive work of the authors. All language recommendations generated by the LLM were rigorously scrutinized and selectively incorporated at the authors' discretion to uphold the accuracy, originality, and integrity of the research. Consequently, the authors bear sole responsibility for the content and findings presented herein, and the LLM does not meet the criteria for authorship or contributorship.

---

**Algorithm 2:** Decomposed Policy Optimization with Importance-weighting (DePO-IW)

---

**Input:** Initial policy parameters $\theta_{\text{init}}$, dataset $\mathcal{D}$, learning rate $\eta$, group size $G$, distillation weight $\lambda$, batch size $B$

**Output:** Optimized policy parameters $\theta$

**1** Initialize policy parameters by $\theta \leftarrow \theta_{\text{init}}$

**2 for** *each training step* **do**

    `// Sample a mini-batch of data`

**3**    Sample a batch $\mathcal{B} = \{(c_{\text{short}}^{(j)}, c_{\text{long}}^{(j)}, x^{(j)}, y^{(j)})\}_{j=1}^{B}$ from $\mathcal{D}$

**4**    Initialize batch losses $\mathcal{L}_{\text{PX}} \leftarrow 0$, $\mathcal{L}_{\text{KD}} \leftarrow 0$

**5**    **for** *each data point $j \in \{1, ..., B\}$ in parallel* **do**

        `// Part 1: Reinforcing Short-context Reasoning as Proxy`
        `   Objective`

        `// Generate rollouts using Short Context`

**6**        Generate $G$ rollouts $\{o_{j,i}\}_{i=1}^{G}$ using $\pi_\theta(\cdot \mid x^{(j)}, c_{\text{short}}^{(j)})$

**7**        Compute rewards $\{r_{j,i}\}_{i=1}^{G}$ where $r_{j,i} = \text{EM}(o_{j,i}, y^{(j)})$

**8**        Compute advantages $\{A_{j,i}\}_{i=1}^{G}$ (normalized within group $j$)

        `// Calculate proxy policy loss`

**9**        Compute proxy loss $\mathcal{L}_{\text{PX}}^{(j)}(\theta)$ using Eq. 1 or Eq. 3

**10**        $\mathcal{L}_{\text{PX}} \leftarrow \mathcal{L}_{\text{PX}} + \mathcal{L}_{\text{PX}}^{(j)}(\theta)$

        `// Part 2: Empower Long-context Reasoning via Knowledge`
        `   Distillation`

        `// Compute importance weights for each rollout`

**11**        **for** *each rollout $i \in \{1, ..., G\}$* **do**

            `// Compute importance weights for each rollout`

**12**            **if** *Token-level weighting* **then**

                `// token-level importance weight can be formulated as:`

**13**                $\widetilde{w}_{j,i,t}(\theta) = \frac{\pi_\theta(o_{j,i,t} \mid x^{(j)}, c_{\text{long}}^{(j)}, o_{j,i,<t})}{\pi_\theta(o_{j,i,t} \mid x^{(j)}, c_{\text{short}}^{(j)}, o_{j,i,<t})}$

**14**            **else if** *Sequence-level weighting* **then**

                `// sequence-level importance weight can be formulated as:`

**15**                $\widetilde{s}_{j,i}(\theta) = \exp\left(\frac{1}{|o_{j,i}|} \sum_{t=1}^{|o_{j,i}|} \log \frac{\pi_\theta(o_{j,i} \mid x^{(j)}, c_{\text{long}}^{(j)}, o_{j,i})}{\pi_\theta(o_{j,i} \mid x^{(j)}, c_{\text{short}}^{(j)}, o_{j,i})}\right)$

**16**            **end**

**17**        **end**

        `// Calculate knowledge distillation loss`

**18**        Compute $\mathcal{L}_{\text{KD}}^{(j)}(\theta)$ by applying weights ($\widetilde{w}_{j,i,t}$ or $\widetilde{s}_{j,i}(\theta)$) to Eq. 1 or Eq. 3

**19**        $\mathcal{L}_{\text{KD}} \leftarrow \mathcal{L}_{\text{KD}} + \mathcal{L}_{\text{KD}}^{(j)}(\theta)$

**20**    **end**

    `// Combine losses and update policy once per batch`

**21**    Compute average compound loss: $\mathcal{L}(\theta) \leftarrow \frac{1}{B}(\mathcal{L}_{\text{PX}} + \lambda \mathcal{L}_{\text{KD}})$

**22**    Update policy parameters: $\theta \leftarrow \theta - \eta \nabla_\theta \mathcal{L}(\theta)$

**23 end**

---

