# OpenReview forum: "Decomposing Policy Optimization into Proxy Objective and Knowledge Distillation for Long-context Reasoning"
_ICLR.cc/2026/Conference — Submitted to ICLR 2026_

### Official Review · Reviewer_szUZ · 2025-10-30

**Soundness:** 3
**Presentation:** 3
**Contribution:** 3
**Rating:** 6
**Confidence:** 4

**Summary:**

This paper explores the problem of long in-context reasoning with reinforcement learning. Existing RL for long-context settings is challenging, given the high computational cost and deteriorating quality of the sampling reasoning trajectories. In this paper, the authors propose a new method called DePO, which decouples policy optimization for long-context reasoning into two parts: (1) first optimize the short-context reasoning patterns via RL; and (2) transfer the short-context reasoning to long-context reasoning through knowledge distillation. Empirical evaluations across multiple benchmarks show the effectiveness of the proposed DePO method.

**Strengths:**

1. The paper is nicely written and very easy to follow.
2. The method proposed in this paper is sound and well-motivated.
3. The authors conduct extensive experiments to demonstrate the effectiveness of the proposed method and obtain in-depth insights.

**Weaknesses:**

1. It is not straightforward for me to directly distill the desired rollout for short context to the long context ones. There are two reasons:
- The answer to the short context is off-policy to the long-context ones.
- There can be an information mismatch. For example, in the long context, there will be passages which does not appear in the short context. Directly training on the short-context answer will lead to insufficient reasoning about the passages which does not appear in the short context, but in the long context.

2. Some experimental settings are not very clear. For example, how to develop the labeled data for the supervised finetuning baselines and preference optimization baselines?

**Questions:**

1. How to solve the insufficient answer problem if we do distillation from the short context answer to the long context answer?
2. How to develop the labeled data for the supervised finetuning baselines and preference optimization baselines? Is it a fair comparison?

---

> ### Author Response · Authors · 2025-11-22
> **Response to Reviewer szUZ (1/3)**
>
> > **W1: It is not straightforward for me to directly distill the desired rollout for short context to the long context ones.**
> >
> > **Q1: How to solve the insufficient answer problem if we do distillation from the short context answer to the long context answer?**
>
> **A1:** We sincerely thank the reviewer for this insightful question. The premise that reasoning skills learned in short contexts can be generalized to longer ones is the cornerstone of DePO, and we appreciate the opportunity to provide a more thorough motivation.
>
> **1. Conceptual Framework and Core Principles**
>
> Our training paradigm synthesizes two core principles: the intrinsic invariance of reasoning patterns and the extrinsic control of the data scaffold.
>
> **1.1 Invariance of Reasoning Patterns Across Context Lengths**
>
> Qualitatively, we observe that the fundamental reasoning patterns a model employs are conserved across varying context lengths. The model's reasoning trajectory can typically be decomposed into three distinct parts: (1)  **Sub-goal Decomposition**, where the main question is broken down into smaller and manageable queries; (2)  **Contextual Grounding**, where the model locates and extracts specific evidence from the text to answer each sub-query; and (3)  **Reasoning Synthesis**, where the extracted evidence is pieced together to form a conclusion.
>
> To illustrate this, consider the reasoning trajectory of the Qwen2.5-7B-Instruct model when answering the question, "How many square miles in the city where KRMP is located?". In the short-context setting, the model produces the following structured response:
>
> (Sub-goal Decomposition) `To determine how many square miles the city where KRMP is located covers, we need to identify the city from the given information about KRMP.`
>
> (Contextual Grounding) `From the document about KRMP, it states: "KRMP (1140 AM) is an urban adult contemporary radio station in Oklahoma City, Oklahoma."`
>
> (Reasoning Summary) `This sentence clearly indicates that KRMP is located in Oklahoma City.`
>
> (Sub-goal Decomposition) `Next, we need to find out the total area of Oklahoma City.`
>
> (Contextual Grounding) `From the document about Oklahoma City, it states: "According to the United States Census Bureau, the city has a total area of 620.34 square miles (1,606.7 km2)..."`
>
> (Reasoning Summary) `Therefore, the answer is 620.34 square miles.`
>
> In the long-context setting, the model produces the following structured response:
>
> (Sub-goal Decomposition) `To determine how many square miles the city where KRMP is located covers, we need to identify the city associated with KRMP from the provided information.`
>
> (Contextual Grounding) `From the document, we find: "KRMP (1140 AM) is an urban adult contemporary radio station in Oklahoma City, Oklahoma."`
>
> (Reasoning Summary) `This tells us that KRMP is located in Oklahoma City, Oklahoma.`
>
> (Contextual Grounding) `Next, we need to find the total area of Oklahoma City. From the information provided:"According to the United States Census Bureau, the city has a total area of 620.34 square miles (1,606.7 km2), of which, 601.11 square miles (1,556.9 km2) of it is land and 19.23 square miles (49.8 km2) of it is water."`
>
> (Reasoning Summary) `Therefore, the total area of the city where KRMP is located (Oklahoma City) is 620.34 square miles.`
>
> This comparison highlights that the fundamental, step-by-step reasoning process is invariant to context length. The difference in performance stems from the increased difficulty of locating evidence and the degradation of reasoning abilities in longer contexts. DePO is designed to capitalize on this invariance. By training on short contexts, we enable the model to learn these core reasoning structures in a clean environment, free from the noise of extensive documents.
>
> **1.2 Isolating Reasoning from Distraction via Contextual Scaffolding**
>
> To facilitate the transfer of reasoning abilities acquired in short contexts to their application in long contexts, we employ a deliberate data construction strategy that serves as a form of contextual scaffolding.
>
> * **Part 1: Short-context Environment Construction.** Following RULER[1], we begin with the set of golden support documents essential for answering a given question. We then construct the short context by augmenting this essential set with a controlled number of irrelevant distractor documents until a moderate length limit is reached. Then, the order of all documents within this context is then randomized.
> * **Part 2: Long-context Environment Construction.** The long context is constructed as a superset of the short context. We take the total documents from $c_\text{short}$ (both support and distractor) and append additional distractor documents to reach much longer context length. The order of documents is also randomized.
>
> Based on this, we ensure that the fundamental evidence and the corresponding reasoning structure remain invariant across different context instances.

---

> ### Author Response · Authors · 2025-11-22
> **Response to Reviewer szUZ (2/3)**
>
> **2. Empirical Support from Training Dynamics:**
>
> Beyond this conceptual framework, our empirical results provide strong quantitative evidence for this generalization. As shown in Fig. 4, which analyzes the training dynamics of our off-policy RL variant (DePO-IW), the policy gradient clipping fraction remains consistently low throughout training. In off-policy RL, a low clipping ratio is a strong indicator that the trajectories sampled from the proxy policy (the policy conditioned on $c_\text{short}$) are highly compatible with the target policy's objective (the policy conditioned on $c_\text{long}$).
>
> **3. Established Precedent for Short-to-Long Generalization**
>
> In addition, the paradigm of short-to-long generalization is gaining significant traction, with recent works like LongPO[2] and SoLoPO[3] demonstrating its effectiveness in preference optimization. These methods validate the core idea that capabilities learned from shorter inputs can be transferred to improve long-context performance. However, extending this paradigm to online RL remains largely unexplored, which involves active trajectory sampling and dynamic policy updates. DePO is designed to fill this specific gap, offers a sound and effective framework.
>
> > **W2:** Some experimental settings are not very clear. For example, how to develop the labeled data for the supervised finetuning baselines and preference optimization baselines?
> >
> > **Q2:** How to develop the labeled data for the supervised finetuning baselines and preference optimization baselines? Is it a fair comparison?
>
> **A2:** We sincerely thank the reviewer for this insightful question regarding the experimental settings for our baselines. This is a crucial point for ensuring the validity and fairness of our comparisons, and we appreciate the opportunity to provide a detailed clarification.
>
> For our evaluation of SFT and PO baselines, we opted to use the official, author-released datasets for training rather than attempting to replicate their complex data synthesis procedures. A key reason is that many pioneering long-context baselines are data-centric, with their novel data generation strategy constituting a major part of their contribution. By using the original curated data, we isolate the methodological differences and ensure that our comparison of DePO against these baselines is both robust and faithful to the original work.
>
> Below, we detail the data construction methodologies for each baseline:
>
> - **LongMIT**[4]: This paper presents the Multi-agent Interactive Multi-hop Generation framework, designed to construct a high-quality instruction dataset for multi-hop reasoning. The framework employs specialized agents to generate foundational single-hop question-answer pairs from a diverse corpus. These pairs are subsequently sampled based on semantic relevance and merged to form complex multi-hop queries requiring information synthesis. To ensure high fidelity, this work introduces a dedicated Quality Verification Agent to evaluate the generated content using a scoring mechanism and filters out suboptimal samples.
> - **LongReward**[5]: The data construction involves a two-stage process for creating datasets for Supervised SFT and DPO. The SFT dataset is initiated by generating diverse question-answering pairs from 10,000 long-context documents via the Self-Instruct technique, using the GLM-4 pre-training corpus and model. This is subsequently augmented with 76k instances from ShareGPT. For preference optimization process, the dataset is constructed by sampling ten responses per prompt. These responses are then scored using GLM-4 as the judge, to form preference pairs by selecting the highest- and lowest-scoring responses for training.
> - **SeaLong-SFT**[6]: SEALONG synthesizes training data via a self-supervised pipeline. The process commences with query-document pairs from the MuSiQue training set, where long contexts are constructed by augmenting relevant documents with randomly sampled irrelevant ones. For each instance, the model generates a candidate pool of multiple reasoning trajectories using temperature sampling. The core of the method lies in a self-evaluation mechanism operationalized through Minimum Bayes Risk, where each candidate is scored based on its semantic consistency with the others in the pool, as measured by sentence embedding similarity. This consensus-based scoring identifies the most plausible reasoning paths, which are then used to create supervision data. The final data can be formatted either as single high-scoring exemplars for supervised fine-tuning or as preference pairs that contrast high-scoring and low-scoring outputs for preference optimization.

---

> ### Author Response · Authors · 2025-11-22
> **Response to Reviewer szUZ (3/3)**
>
> - **Pos2Distill-R$^2$**[7]: The data construction process begins by identifying an advantageous position, which empirically determined to be the recent slots in the context. The base model is then prompted with the relevant documents placed in this advantageous configuration to generate high-quality reasoning trajectory. This response serves as teacher output for training. Subsequently, for the same query, multiple trivial input prompts are constructed. In these prompts, the same set of critical documents are systematically placed at various other positions within the context window, which are known to induce performance degradation. Each of these unfavorable prompts is then paired with the high-quality reasoning trajectory generated from the advantageous position.
> - **LongPO**: The procedure begins by curating a long-context corpus from existing sources. For a given long document, the core of the method involves a reverse-generation process. First, a shorter self-contained chunk of text is randomly sampled from the long document. It uses self-Instruct methodology to prompt well-aligned short-context language model to generate a relevant instruction based on this short chunk, ensuring the chunk contains all necessary information to answer the instruction. Subsequently, this same model generates two responses to the instruction: a "chosen" response conditioned on the short relevant chunk, and a "rejected" response conditioned on the entire long document. This creates a preference pair that captures the performance discrepancy between the model's well-aligned short-context capabilities and its under-aligned long-context behavior.
> - **SoLoPO**: The process begins with the MuSiQue multi-hop question-answering dataset. Following the context synthesis strategy from the RULER, for each question and its corresponding supporting documents, both a short context and a long context are generated. This is achieved by combining the essential supporting documents with a varying number of randomly sampled, irrelevant documents to reach the target lengths, thereby simulating the information redundancy common in real-world long texts. Subsequently, preference pairs are generated for these contexts. These generated responses are then evaluated against the answers to identify the highest-quality response as the chosen answer and a lower-quality one as the rejected answer.
> - **LongFaith**[8]: For the SFT dataset, the process begins by using a LLM to synthesize faithful reasoning chains. This is achieved by providing the model with a question, the answer, and the specific supporting facts from the source documents, using a chain-of-citation prompt that requires the model to provide attributions for each reasoning step. To create the preference optimization dataset, this high-faithfulness reasoning chain serves as the chosen sample, which is then contrasted with rejected samples exhibiting questionable faithfulness. These rejected samples are intentionally synthesized to model three specific failure modes, includes misinformation, lack of attribution, and potential knowledge conflicts.
>
> [1] RULER: What's the Real Context Size of Your Long-Context Language Models? COLM, 2024.
>
> [2] LongPO: Long context self-evolution of large language models through short-to-long preference optimization. ICLR, 2025.
>
> [3] SoLoPO: Unlocking Long-Context Capabilities in LLMs via Short-to-Long Preference Optimization. arXiv, 2025.
>
> [4] What are the essential factors in crafting effective long context multi-hop instruction datasets? insights and best practices. ACL, 2025.
>
> [5] Longreward: Improving long-context large language models with ai feedback. ACL, 2025.
>
> [6] Large Language Models Can Self-Improve in Long-context Reasoning. arXiv, 2025.
>
> [7] Position Bias Mitigates Position Bias:Mitigate Position Bias Through Inter-Position Knowledge Distillation. EMNLP, 2025.
>
> [8] LongFaith: Enhancing Long-Context Reasoning in LLMs with Faithful Synthetic Data. ACL Findings, 2025.
>
> ---
>
> We thank you again for your valuable feedback and sincerely hope this response addresses your concerns. We would be pleased to engage in any further discussion.

---

> ### Author Response · Authors · 2025-11-26
> **Gratitude and Awaiting Your Feedback**
>
> Dear Reviewer szUZ,
>
> Thank you once again for your time and for your valuable feedback on our manuscript.
>
> We have carefully considered all of your comments and have provided a comprehensive response. The corresponding revisions have been made in the manuscript, and we believe these changes have significantly improved our work.
>
> We sincerely hope that our responses and the revised manuscript will resolve your concerns. We are happy to engage in any further discussion.
>
> Best regards,
>
> Authors

---

### Official Review · Reviewer_b3vd · 2025-10-31

**Soundness:** 2
**Presentation:** 3
**Contribution:** 3
**Rating:** 4
**Confidence:** 3

**Summary:**

The paper introduces **DePO**, a novel reinforcement learning framework that addresses the inefficiency of training large language models (LLMs) on long-context tasks. DePO **decouples policy optimization** into two parallel components: (1) reinforcement learning on short-context reasoning as a **proxy objective** to efficiently learn high-quality reasoning behaviors, and (2) **knowledge distillation** to transfer these learned reasoning patterns to long-context settings.

This design mitigates the high computational cost and trajectory quality degradation commonly encountered in long-context RL, resulting in reduced training time and improved accuracy compared to direct long-context RL methods. DePO generalizes across different RL algorithms (e.g., GRPO, DAPO, GSPO) and foundation models (Qwen, Llama), maintaining strong short-context reasoning performance while enhancing long-context reasoning.

**Strengths:**

1. **Strong Experiment Results** Experiments across multiple long-document QA benchmarks show that DePO consistently outperforms supervised fine-tuning, preference optimization, and direct long-context RL methods, achieving 2.8% higher accuracy and 49.5% less training time.

2. **Compatibility**: DePO proves compatible with multiple RL algorithms (GRPO, DAPO, GSPO) and different foundation models (Qwen and Llama series), maintaining consistent performance gains and scalability across architectures and training paradigms

**Weaknesses:**

1. **Methodological Clarity** I think the methodology section lacks clarity to some extent. Regarding the knowledge distillation, I am under the impression that the teacher model and student model are being updated simultaneously. Is this the correct understanding of the proposed approach? The teacher is defined as the policy conditioned on the short context, and the student model refers to the same policy conditioned on the long context (line 251). Equation 5 in line 298 indicates that the short context policy model (teacher) and knowledge distillation training (student) are being trained jointly. If this is the case, then there is a concern for model collapse.

2. **Reward Sparsity Concern** Using exact match as a reward signal may lead to reward sparsity issue, something not adequately addressed by the paper.

**Questions:**

1. Are the teacher model and student model being updated simultaneously?

2. Since the overall performance gain is very small, I am wondering if there is any fluctuation of performance across multiple runs? Did you perform any significance test?

---

> ### Author Response · Authors · 2025-11-22
> **Response to Reviewer b3vd (1/2)**
>
> > **W1:** Methodological Clarity I think the methodology section lacks clarity to some extent. Regarding the knowledge distillation, I am under the impression that the teacher model and student model are being updated simultaneously. Is this the correct understanding of the proposed approach? The teacher is defined as the policy conditioned on the short context, and the student model refers to the same policy conditioned on the long context (line 251). Equation 5 in line 298 indicates that the short context policy model (teacher) and knowledge distillation training (student) are being trained jointly. If this is the case, then there is a concern for model collapse.
> >
> > **Q1:** Are the teacher model and student model being updated simultaneously?
>
> **A1:** We sincerely thank you for your insightful review, and we appreciate the opportunity to provide a more detailed explanation.
>
> **1. On the Simultaneous Update of the Teacher and Student**
>
> As observerd by the reviewer, we do not use two independent policy models. Instead, a **single policy model, $\pi_\theta$,** serves as both the teacher model and the student model.
>
> * The teacher model is the policy model $\pi_\theta$ operating under the condition of a short context $c_\text{short}$.
> * The student model is the same policy model $\pi_\theta$ operating under the condition of a long context $c_\text{long}$.
>
> The reinforcement learning objective $J_\text{PX}(\theta)$ (for updating the teacher) and the knowledge distillation objective $J_\text{KD}(\theta)$ (for aligning the student with the teacher) are jointly optimized via the combined objective in Eq. 5.
>
> **2. Why DePO Avoids Model Collapse**
>
> Model collapse could indeed occur if the teacher policy were to degenerate or drift arbitrarily. However, our DePO framework is designed to prevent this through the synergistic interplay of its two loss components:
>
> * **Proxy Objective $\mathcal{J}_\text{PX}$ as an Anchor:** This component is a standard reinforcement learning objective that is optimized exclusively within the short-context environment. As we demonstrate in Fig. 1, the model exhibits superior performance and reasoning quality in the short-context setting. The RL process provides a clear external signal through verifiable rewards, continually pushing the teacher policy $\pi_\theta(\cdot| c_\text{short}, x)$ to improve its reasoning capabilities and achieve higher rewards. This objective serves as a stable anchor, ensuring that the teacher's policy is not an arbitrarily drifting target, but one that is actively guided towards high-quality and high-reward states.
> * **Knowledge Distillation ($J_\text{KD}$) as a Generalization Bridge:** The $J_\text{KD}(\theta)$ term transfers the robust reasoning skills learned in the simpler short-context environment to the more challenging long-context setting. It essentially regularizes the model's behavior in long contexts, encouraging it to emulate its own behavior from the short contexts.
>
> In summary, the joint optimization in DePO is not a process where two parts of the model are blindly chasing each other. The external rewards and anchoring mechanism prevent the system from collapsing into a meaningless state. Our experimental results in Table 2 and Fig. 3, which show consistent performance gains over baselines, further validate that our approach successfully enhances long-context reasoning rather than leading to model collapse.
>
> > **W2: Reward Sparsity Concern Using exact match as a reward signal may lead to reward sparsity issue, something not adequately addressed by the paper.**
>
> **A2:** Thank you for your insightful comment. Our decision to employ EM was a deliberate trade-off, made after a thorough evaluation of several reward mechanisms.
>
> While recent work has explored combining SubEM with LLM-as-a-judge, this approach remains susceptible to reward hacking. Specifically, the nature of SubEM can incentivize the model to generate over long responses to maximize its score by including as many keywords as possible. This behavior severely compromises the precision and conciseness of the answers.
>
> In contrast, although the EM signal is sparse, it provides a clear, unambiguous, and verifiable supervisory signal that effectively guides the model's optimization. Our experimental results demonstrate that, the reinforcement learning process can achieve strong performance by leveraging this sparse yet reliable signal.
>
> Furthermore, adopting EM as a binary reward greatly simplifies the hyperparameter tuning process. In our DePO, we can simply choose the reward to 1 as high-quality trajectory.

---

> ### Author Response · Authors · 2025-11-22
> **Response to Reviewer b3vd (2/2)**
>
> > **Q2:** Since the overall performance gain is very small, I am wondering if there is any fluctuation of performance across multiple runs? Did you perform any significance test?
>
> **A3:** To mitigate performance fluctuations and ensure reproducibility, our experimental implementation adhered to the configurations of prior studies, setting the temperature to 0. We posit that the performance gains achieved by our method are significant. As demonstrated in Table 2, DePO outperforms SFT-based methods by 7.7%, surpasses PO methods by 12.7%, and exceeds RL-based methods by 2.8%. Furthermore, as shown in Table 3, DePO yields consistent improvements when integrated with various reinforcement learning algorithms or different base models.
>
> To further validate that these improvements are robust and statistically significant, we replicated our main experiments across three different random runs. The subsequent table presents a comparative analysis of the mean performance between our method (DePO) and DAPO (Long) on all long-context benchmarks. The outcomes are reported as mean $\pm$ standard deviation, with corresponding p-values from paired t-tests provided to substantiate the statistical significance of our findings. The resulting p-value is 0.009.
>
> | Methods          | 2Wiki        | HQA          | Musi         | DocMath      | Frames       | 2Wiki       | HQA          | Musi         | NarQA        | Qasp         | Avg.         |
> | ---------------- | ------------ | ------------ | ------------ | ------------ | ------------ | ----------- | ------------ | ------------ | ------------ | ------------ | ------------ |
> | DAPO (Long)      | 69.9 $\pm$ 2.6 | 72.0 $\pm$ 1.5 | 52.4 $\pm$ 1.7 | 40.5 $\pm$ 1.3 | 42.0 $\pm$ 0.5  | 66.2 $\pm$ 1.8 | 59.9 $\pm$ 2.3 | 46.3 $\pm$ 0.7 | 25.9 $\pm$ 1.3 | 37.9 $\pm$ 4.3 | 51.3 $\pm$ 1.0 |
> | DePO$^\dagger$ | 74.2 $\pm$ 0.5  | 74.2 $\pm$ 0.1  | 58.4 $\pm$ 0.8  | 42.0 $\pm$ 0.9  | 42.1 $\pm$ 0.8 | 70.8 $\pm$ 2.5 | 59.0 $\pm$ 1.1  | 51.1 $\pm$ 1.6 | 26.9 $\pm$ 0.2 | 43.6 $\pm$ 0.4 | 54.3 $\pm$ 0.4  |
>
> ---
>
> We thank you again for your valuable feedback and sincerely hope this response addresses your concerns. We would be pleased to engage in any further discussion.

---

> ### Author Response · Authors · 2025-11-26
> **Gratitude and Awaiting Your Feedback**
>
> Dear Reviewer b3vd,
>
> Thank you once again for your time and for your valuable feedback on our manuscript.
>
> We have carefully considered all of your comments and have provided a comprehensive response. The corresponding revisions have been made in the manuscript, and we believe these changes have significantly improved our work.
>
> We sincerely hope that our responses and the revised manuscript will resolve your concerns. We are happy to engage in any further discussion.
>
> Best regards,
>
> Authors

---

### Official Review · Reviewer_8Zdd · 2025-10-31

**Soundness:** 4
**Presentation:** 4
**Contribution:** 3
**Rating:** 6
**Confidence:** 4

**Summary:**

This paper proposes a novel approach to enhancing the long-context capabilities of large language models (LLMs) through reinforcement learning. The authors design a clever setup that constructs both short-context and long-context versions of documents for a given query. The policy is first trained on short-context data and then used as a teacher to guide training on long-context documents. The idea is interesting, the writing is clear, and the overall presentation is well-organized and visually appealing.

The main concern lies in the generalizability of the proposed method for long-context training. The approach assumes that for every long-context input, a corresponding short-context version can be found, which may hold true for retrieval-augmented generation (RAG) tasks but not necessarily for other long-context scenarios. I would be interested in hearing the authors’ thoughts on how their algorithm could be extended to handle more complex, non-RAG settings.

Overall, the idea is novel and promising, and I would recommend the paper for acceptance.

**Strengths:**

- The paper proposes a novel and well-motivated approach to enhance the long-context capabilities of LLMs using reinforcement learning.
- The idea of constructing paired short-context and long-context documents for the same query is clever and intuitively appealing. And the two-stage training strategy based on the constructed data — first training on short-context data and then transferring to long-context settings — is conceptually sound and easy to follow.
- The writing and presentation quality are excellent; the paper is clearly structured, visually appealing, and easy to read.
- The experimental setup and motivation are coherent, making the contributions easy to understand and evaluate.

**Weaknesses:**

- The generalizability of the proposed approach is limited. The method assumes that for every long-context input, there exists a corresponding short-context version, which may hold for RAG-style tasks but not for other long-context scenarios such as reasoning or summarization.

- The paper lacks a discussion or analysis of how the proposed algorithm could be extended or adapted to non-RAG settings, where short-context counterparts are not naturally available.

**Questions:**

- How can this method generalize to non-RAG settings where high-quality short-context inputs cannot be easily obtained for each long-context instance?

- Some clarification is needed regarding the different variants of DePO (GROP, DAPO, GSPO). In Table 2, is the reported result based on DAPO, while Table 3 combines DAPO and GSPO? And are all later tables and figures based on GROP?

- In Figure 5, how is the training time comparison between DePO and GROP conducted? Did you fix the total number of training steps, set a performance threshold and measure the steps needed to reach it, or use another criterion?

- The performance visualization in Figure 5 seems slightly misleading — the reported improvement is only about $5.3\%$, yet the blue and orange bars appear to differ by nearly a factor of two. This is not a major issue, but clarification or a note on scaling would be appreciated.

- The description of the DePO-IW algorithm in the appendix needs further clarification. It appears that this algorithm computes a knowledge-distillation loss rather than a KL divergence between student and teacher policies, and that it applies the policy gradient twice — with the second application (line 16 in Algorithm 2) being somewhat off-policy with reweighting. I am not sure whether I understand correctly.

---

> ### Author Response · Authors · 2025-11-22
> **Response to Reviewer 8Zdd (1/2)**
>
> > **W1:** The generalizability of the proposed approach is limited. The method assumes that for every long-context input, there exists a corresponding short-context version, which may hold for RAG-style tasks but not for other long-context scenarios such as reasoning or summarization.
> >
> > **W2:** The paper lacks a discussion or analysis of how the proposed algorithm could be extended or adapted to non-RAG settings, where short-context counterparts are not naturally available.
> >
> > **Q1:** How can this method generalize to non-RAG settings where high-quality short-context inputs cannot be easily obtained for each long-context instance?
>
> **A1:** We thank the reviewer for raising this important point. Our training objective is to enhance the model's intrinsic long-context capabilities, and this has been thoroughly validated on various long-document reasoning benchmarks in Table 2 and Table 3.
>
> To demonstrate the generalizability of our method beyond RAG scenarios, we evaluated DePO$^\dagger$ on a suite of benchmarks, including GovReport [1], QMSum [2], and MultiNews [3] for summarization, and TriviaQA [4], SAMSum [5], TREC [6], and LSHT [7] for in-context learning. As shown in the table below, DePO$^\dagger$ achieves consistently strong performance across these diverse tasks.
>
> | Method              | GovReport | QMSum | MultiNews | TriviaQA | SAMSum | TREC | LSHT |Avg.|
> | ------------------- | --------- | ----- | --------- | -------- | ------ | ---- | ---- | ---- |
> | Qwen2.5-7B-Instruct | 33.5      | 24.2  | 25.2      | 88.1     | 40.5   | 51.5 | 35.3 | 42.6 |
> | DePO$^\dagger$     | 34.0      | 24.5  | 25.5      | 90.4     | 42.4   | 64.5 | 40.5 | 46.0 |
>
> While applying our framework directly to information-dense long-context tasks presents challenges, we believe the underlying decomposed optimization paradigm of DePO holds significant potential for a broad range of long-context related tasks. For instance:
>
> * **Repository-level code completion:** The short-context could be function signatures and key dependencies, while the long-context is the entire repository.
> * **Object detection in computer vision:** The short-context could be a cropped image containing only the target object, while the long-context is the complex original scene containing that object.
>
> As mentioned in the conclusion, we will extend the paper to include a discussion on potential application scenarios and outline corresponding adaptation strategies, positioning them as promising directions for future research.
>
> > **Q2:** Some clarification is needed regarding the different variants of DePO (GRPO, DAPO, GSPO). In Table 2, is the reported result based on DAPO, while Table 3 combines DAPO and GSPO? And are all later tables and figures based on GROP?
>
> **A2:** We sincerely thank the reviewer for raising this important point. We acknowledge that our manuscript did not adequately distinguish which variant of our method (GRPO, DAPO, GSPO) was used for each result, and we appreciate the opportunity to clarify this.
>
> The results in Table 2 were generated using our method integrated with DAPO, the results in Table 3 demonstrate the integration with DAPO and GSPO, and the results in Table 4, Table 5, Fig. 5, Fig. 6 utilize GRPO.
>
> In our revision, we take concrete steps to eliminate this ambiguity. We will add a clear statement to the caption of each table and figure specifying the variant used. For Table 3, we will adopt the following precise notation to make the distinction unambiguous: DePO$^\dagger$ will denote the integration with DAPO, and DePO$^\ddagger$ will denote the integration with GSPO.
>
> > **Q3:** In Figure 5, how is the training time comparison between DePO and GRPO conducted? Did you fix the total number of training steps, set a performance threshold and measure the steps needed to reach it, or use another criterion?
>
> **A3:** Thank you for your insightful question.
> To ensure a fair and direct comparison, we fixed the total number of training steps for both DePO and the GRPO baseline. The Training Time reported in Fig. 5 (right y-axis) represents the time required for one step within the training process for each method, measured under different context lengths. For the performance evaluation (left y-axis), we selected the final model for each method by choosing the checkpoint that achieved the best performance on our validation set within this fixed training schedule.

---

> > ### Author Response · Authors · 2025-11-22
> > **Response to Reviewer 8Zdd (2/2)**
> >
> > > **Q4:** The performance visualization in Figure 5 seems slightly misleading — the reported improvement is only about , yet the blue and orange bars appear to differ by nearly a factor of two. This is not a major issue, but clarification or a note on scaling would be appreciated.
> >
> > **A4:** We sincerely thank you for your valuable feedback. We apologize for the potential for misinterpretation.
> >
> > Our initial scaling was chosen to emphasize performance nuances. To address this, we have adjusted the figure in our revised manuscript. The new Y-axis range mitigates the visual distortion while still effectively showcasing the performance differences between methods.
> >
> > > **Q5:** The description of the DePO-IW algorithm in the appendix needs further clarification. It appears that this algorithm computes a knowledge-distillation loss rather than a KL divergence between student and teacher policies, and that it applies the policy gradient twice — with the second application (line 16 in Algorithm 2) being somewhat off-policy with reweighting. I am not sure whether I understand correctly.
> >
> > **A5:** We sincerely thank the reviewer for insightful question.
> >
> > **1. On Knowledge Distillation vs. Off-Policy Policy Gradient:**
> >
> > In our DePO-IW, we reframe the knowledge distillation as an off-policy reinforcement learning problem. Instead of using KL divergence to directly align the output distributions, DePO-IW achieves this by using the high-quality trajectories sampled from the short-context policy as off-policy data to update the long-context target policy.
> >
> > **2. On Applying Policy Gradient Twice:**
> >
> > The policy gradient is intentionally applied twice: first to the short-context proxy policy, and second to the long-context  target policy.
> >
> > * **Proxy Policy**:  This corresponds to Line 9 in Algorithm 2. It is an on-policy policy gradient update. Its goal is to improve the policy's reasoning ability within the tractable short-context environment. The trajectories are sampled from $\pi(\cdot|c_\text{short}, x)$, and the policy is updated to maximize the expected rewards in this same short-context setting. This step is responsible for learning effective reasoning patterns.
> > * **Target Policy**: This corresponds to Line 16 in Algorithm 2. It is an off-policy policy gradient update with importance sampling reweighting.
> >   * It reuses the same trajectories and their associated rewards, which were sampled from the short-context policy.
> >   * The importance weights correct for the distributional shift. This update encourages the long-context policy to produce the same high-reward trajectories that the short-context policy discovered.
> >
> > In summary, DePO-IW's two policy gradient updates are all essential. By leveraging off-policy data, we avoid costly and often uninformative exploration in the long-context environment. The superior performance of DePO-IW, as demonstrated in Table 3, empirically validates the effectiveness of this approach.
> >
> > [1] Efficient attentions for long document summarization. NAACL, 2021.
> >
> > [2] QMSum: A new benchmark for query-based multi-domain meeting summarization. NAACL, 2021.
> >
> > [3] Multi-news: A large-scale multi-document summarization dataset and abstractive hierarchical model. ACL, 2019.
> >
> > [4] Triviaqa: A large scale distantly supervised challenge dataset for reading comprehension. ACL, 2017.
> >
> > [5] SAMSum corpus: A human-annotated dialogue dataset for abstractive summarization. ACL, 2019.
> >
> > [6] Task definition for large scale text categorization at nlpcc 2014. NLPCC, 2014.
> >
> > ---
> >
> > We thank you again for your valuable feedback and sincerely hope this response addresses your concerns. We would be pleased to engage in any further discussion.

---

> ### Author Response · Authors · 2025-11-26
> **Gratitude and Awaiting Your Feedback**
>
> Dear Reviewer 8Zdd,
>
> Thank you once again for your time and for your valuable feedback on our manuscript.
>
> We have carefully considered all of your comments and have provided a comprehensive response. The corresponding revisions have been made in the manuscript, and we believe these changes have significantly improved our work.
>
> We sincerely hope that our responses and the revised manuscript will resolve your concerns. We are happy to engage in any further discussion.
>
> Best regards,
>
> Authors

---

### Official Review · Reviewer_oY4b · 2025-11-03

**Soundness:** 3
**Presentation:** 2
**Contribution:** 3
**Rating:** 4
**Confidence:** 2

**Summary:**

This paper proposes Decomposed Policy Optimization (DePO), which decomposes long-context policy optimization for reinforcement learning (RL) into two parts: learning a policy on short-context environments, and then using knowledge distillation to transfer it to long-context environments. The RL model can thus learn robust local reasoning on the short-context datasets. The paper further introduces a variant of DePO, called DePO-IW, that formulates the distillation process as an off-policy reinforcement learning one. The paper presents extensive numerical comparisons to various types of baseline algorithms for long contexts, on ten different datasets, showing that integrating DePO or DePO-IW into existing RL methods generally increases their accuracy and yields higher accuracy than supervised fine-tuning or preference optimization methods.

**Strengths:**

+ The experimental results seem quite thorough, covering ten datasets and multiple supervised fine-tuning, preference optimization, and reinforcement learning baselines. In most of these comparisons, DePO and its variant DePO-IW improve the performance of these baselines.

+ The paper addresses an important problem of handling long contexts, and it can be integrated into existing reinforcement learning methods for long-context queries.

**Weaknesses:**

--The intuition behind why we would expect any reasoning learned on short contexts to be useful for longer contexts is explained only very briefly, with the idea that local reasoning can be learned on short contexts without being confounded by long contexts. Given that this is the entire premise of DePO, I’d expect a more thorough discussion motivating the method and explaining why it would be expected to work.

--The technical challenges of DePO are not explained well. Beyond the insight that we can transfer reasoning learned on short contexts to long contexts, the paper does not seem to have any technical novelty. The reinforcement learning formulations in Equations (3) to (7) are taken from past work, and that in Equation (7) is almost the same as the formulation in Equation (3), yielding little additional insight.

-- Many of the details of knowledge distillation (KD) are missing from the paper. For example, how often is KD performed? Only once, after the reinforcement learning model is trained on short contexts? Can any KD algorithm be used, or does the paper need to adapt existing methods to this setting?

**Questions:**

1) How long does the KD algorithm take to run in practice? Does this represent significant computational overhead compared to training the reinforcement learning methods?

2) In Table 3, neither DePO nor DePO-IW consistently outperform each other. Under which circumstances would a user want to use DePO-IW instead of DePO? Are there particular problem characteristics that might indicate whether one or the other would perform better?

3) Lemma 3.1 does not seem to use any specifics of the long- or short-context settings, and appears to apply to any scenario in which we apply a policy to two different distributions. Does it apply only to the present long vs. short context setting, or can it be applied to these more general settings as well?

Please see also the weaknesses above.

---

> ### Author Response · Authors · 2025-11-22
> **Response to Reviewer oY4b (1/5)**
>
> > **W1:** The intuition behind why we would expect any reasoning learned on short contexts to be useful for longer contexts is explained only very briefly, with the idea that local reasoning can be learned on short contexts without being confounded by long contexts.
>
> **A1:** We sincerely thank the reviewer for this insightful question. The premise that reasoning skills learned in short contexts can be generalized to longer ones is the cornerstone of DePO, and we appreciate the opportunity to provide a more thorough motivation.
>
> **1. Conceptual Framework and Core Principles**
>
> Our training paradigm synthesizes two core principles: the intrinsic invariance of reasoning patterns and the extrinsic control of the data scaffold.
>
> **1.1 Invariance of Reasoning Patterns Across Context Lengths**
>
> Qualitatively, we observe that the fundamental reasoning patterns a model employs are conserved across varying context lengths. The model's reasoning trajectory can typically be decomposed into three distinct parts: (1)  **Sub-goal Decomposition**, where the main question is broken down into smaller and manageable queries; (2)  **Contextual Grounding**, where the model locates and extracts specific evidence from the text to answer each sub-query; and (3)  **Reasoning Synthesis**, where the extracted evidence is pieced together to form a conclusion.
>
> To illustrate this, consider the reasoning trajectory of the Qwen2.5-7B-Instruct model when answering the question, "How many square miles in the city where KRMP is located?". In the short-context setting, the model produces the following structured response:
>
> (Sub-goal Decomposition) `To determine how many square miles the city where KRMP is located covers, we need to identify the city from the given information about KRMP.`
>
> (Contextual Grounding) `From the document about KRMP, it states: "KRMP (1140 AM) is an urban adult contemporary radio station in Oklahoma City, Oklahoma."`
>
> (Reasoning Summary) `This sentence clearly indicates that KRMP is located in Oklahoma City.`
>
> (Sub-goal Decomposition) `Next, we need to find out the total area of Oklahoma City.`
>
> (Contextual Grounding) `From the document about Oklahoma City, it states: "According to the United States Census Bureau, the city has a total area of 620.34 square miles (1,606.7 km2)..."`
>
> (Reasoning Summary) `Therefore, the answer is 620.34 square miles.`
>
> In the long-context setting, the model produces the following structured response:
>
> (Sub-goal Decomposition) `To determine how many square miles the city where KRMP is located covers, we need to identify the city associated with KRMP from the provided information.`
>
> (Contextual Grounding) `From the document, we find: "KRMP (1140 AM) is an urban adult contemporary radio station in Oklahoma City, Oklahoma."`
>
> (Reasoning Summary) `This tells us that KRMP is located in Oklahoma City, Oklahoma.`
>
> (Contextual Grounding) `Next, we need to find the total area of Oklahoma City. From the information provided:"According to the United States Census Bureau, the city has a total area of 620.34 square miles (1,606.7 km2), of which, 601.11 square miles (1,556.9 km2) of it is land and 19.23 square miles (49.8 km2) of it is water."`
>
> (Reasoning Summary) `Therefore, the total area of the city where KRMP is located (Oklahoma City) is 620.34 square miles.`
>
> This comparison highlights that the fundamental, step-by-step reasoning process is invariant to context length. The difference in performance stems from the increased difficulty of locating evidence and the degradation of reasoning abilities in longer contexts. DePO is designed to capitalize on this invariance. By training on short contexts, we enable the model to learn these core reasoning structures in a clean environment, free from the noise of extensive documents.
>
> **1.2 Isolating Reasoning from Distraction via Contextual Scaffolding**
>
> To facilitate the transfer of reasoning abilities acquired in short contexts to their application in long contexts, we employ a deliberate data construction strategy that serves as a form of contextual scaffolding.
>
> * **Part 1: Short-context Environment Construction.** Following RULER[1], we begin with the set of golden support documents essential for answering a given question. We then construct the short context by augmenting this essential set with a controlled number of irrelevant distractor documents until a moderate length limit is reached. Then, the order of all documents within this context is then randomized.
> * **Part 2: Long-context Environment Construction.** The long context is constructed as a superset of the short context. We take the total documents from $c_\text{short}$ (both support and distractor) and append additional distractor documents to reach much longer context length. The order of documents is also randomized.
>
> Based on this, we ensure that the fundamental evidence and the corresponding reasoning structure remain invariant across different context instances.

---

> ### Author Response · Authors · 2025-11-22
> **Response to Reviewer oY4b (2/5)**
>
> **2. Empirical Support from Training Dynamics:**
>
> Beyond this conceptual framework, our empirical results provide strong quantitative evidence for this generalization. As shown in Fig. 4, which analyzes the training dynamics of our off-policy RL variant (DePO-IW), the policy gradient clipping fraction remains consistently low throughout training. In off-policy RL, a low clipping ratio is a strong indicator that the trajectories sampled from the proxy policy (the policy conditioned on $c_\text{short}$) are highly compatible with the target policy's objective (the policy conditioned on $c_\text{long}$).
>
> **3. Established Precedent for Short-to-Long Generalization**
>
> In addition, the paradigm of short-to-long generalization is gaining significant traction, with recent works like LongPO[2] and SoLoPO[3] demonstrating its effectiveness in preference optimization. These methods validate the core idea that capabilities learned from shorter inputs can be transferred to improve long-context performance. However, extending this paradigm to online RL remains largely unexplored, which involves active trajectory sampling and dynamic policy updates. DePO is designed to fill this specific gap, offers a sound and effective framework.
>
> > W2: The technical challenges of DePO are not explained well.
>
> **A2:** Thank you for your constructive feedback. We would like to clarify the technical novelty and challenges addressed by our DePO. We respectively contend that **the core technical novelty of our work lies not in inventing a new loss function, but in the design of the Decomposed Policy Optimization** framework itself. This framework strategically combines reinforcement learning and knowledge distillation to specifically overcome the severe and practical challenges of online RL for long-context LLMs.
>
> **1. The Severe Technical Challenges of Naive Long-Context RL**
>
> We argue that technical difficulty of this problem is non-trivial. Standard online RL methods (e.g., GRPO) are suboptimal for long-context scenarios due to two critical issues, which we empirically demonstrate in Fig. 1:
>
> * **Degradation in Reasoning Trajectory Quality:** As context length increases, the quality of sampled reasoning trajectories deteriorates. These low-quality trajectories produce noisy and ineffective policy gradients, which undermines the stability of the training process.
> * **Prohibitive Computational Cost:** The computational overhead of sampling increases dramatically. This makes direct online RL on long contexts computationally expensive.
>
> Given the foregoing discussion, the direct application of existing RL algorithms to long contexts is therefore not only inefficient but also yields suboptimal results. This is the core technical barrier our work aims to break.
>
> **2. DePO: A Novel Decomposed Optimization Paradigm**
>
> DePO elegantly resolves these challenges by decomposing the complex long-context optimization problem into two more tractable sub-problems: (1) direct optimization within short contexts, and (2) knowledge distillation from short to long contexts, all adeptly situated within a reinforcement learning framework.
>
> * **Proxy Objective**: By constraining online RL to a short-context environment, we leverage the advantages of reasoning trajectory quality, and efficiency inherent to short-context RL. This is a deliberate design choice to ensure that an expert policy learns core reasoning skills effectively, insulated from the noise and complexity of long contexts.
> * **Knowledge Distillation**: This component is designed to extrapolate the reasoning skills learned on short contexts to the target long-context setting. It achieves this by aligning the policy's output distribution under a long context with its high-quality output distribution under a short context. The formulation of $J_\text{KD}$ is a novel application of forward KL-divergence to bridge the generalization gap for context length in online RL. We further provide a direct mathematical justification in Lemma 3.1.
>
> **3. Clarification on the Similarity of Eq. 3 and Eq.7**
>
> We agree that Eq. 7 and Eq. 3 are structurally similar, as they both represent standard policy optimization objectives. However, their operational difference is the cornerstone of our contribution. **The crucial distinction lies in the input context used for sampling and optimization:**
>
> * **Standard RL**: Directly samples and optimizes on long contexts.
> * **DePO's Proxy Objective**: Explicitly restricts sampling and optimization to short contexts. This is precisely where our efficiency and quality gains originate.
>
> This is not a trivial similarity but a fundamental design choice that underpins the entire framework.

---

> ### Author Response · Authors · 2025-11-22
> **Response to Reviewer oY4b (3/5)**
>
> **4. Empirical Validation of Technical Novelty**
>
> The technical value of our framework is validated by its empirical success in overcoming the initial challenges:
>
> * **Superior Performance:** DePO achieves an average improvement of 2.8% over RL methods applied directly to long contexts.
> * **Efficiency Gains:** DePO reduces training time overhead by 49.5% compared to naive long-context GRPO.
> * **General Framework:** As shown in Table 3, DePO consistently improves performance when integrated with different RL algorithms (GRPO, DAPO, GSPO). This demonstrates that the benefit stems from our decomposed paradigm, not from a specific RL algorithm, confirming the novelty and effectiveness of the framework.
>
> In conclusion, DePO's technical contribution is a paradigm shift for long-context RL. It decomposes a complex optimization problem into two manageable objectives, simultaneously addressing the dual challenges of reasoning trajectory quality degradation and high computational cost. We believe this clarification adequately demonstrates DePO's significant technical contribution to the field of long-context reasoning.
>
> > **W3:** Many of the details of knowledge distillation (KD) are missing from the paper. For example, how often is KD performed? Only once, after the reinforcement learning model is trained on short contexts? Can any KD algorithm be used, or does the paper need to adapt existing methods to this setting?
>
> **A3:** We sincerely thank the reviewer for their valuable comment on the clarity of our KD process. We appreciate this opportunity to provide a more detailed explanation.
>
> Crucially, within our DePO framework, KD is **not a separate, post-hoc stage executed "only once" after the RL training is complete**. Instead, DePO integrates RL and KD into a  **simultaneous** **optimization process**. At each update step, we concurrently optimize both the proxy policy objective and the knowledge distillation objective to update the current policy model.
>
> As demonstrated in Eq. 5 and Algorithm 1 (specifically, lines 10-14) of our paper, we simultaneously compute and optimize two objectives in every training iteration:
>
> * **Proxy Objective**: This objective drives the policy update within the short-context environment.
> * **Knowledge Distillation**: This objective serves as an alignment loss to transfer the model's capabilities learned in short contexts to long-context scenarios.
>
> These two losses are combined via a weighted sum to form a unified composite objective function that guides the update of the policy model's parameters $\theta$. Therefore, RL and KD are conducted in parallel and are integral to the entire training process, rather than being two sequential steps.
>
> In this work, we explore and implement two KD paradigms that are compatible with our RL objective:
>
> - **DePO (based on Forward KL-Divergence)**: DePO adopts the classic concept of knowledge distillation. As presented in our KD objective (Eq. 10) in Sec. 3.2.2, this approach aims to minimize the **Forward KL-Divergence** between the output probability distributions of the model under short-context and long-context conditions. This alignment is performed on high-quality trajectories that are conditionally filtered, and the theoretical rationale for this approach is provided by Lemma 3.1. This directly encourages distributional consistency across different context lengths.
> - **DePO-IW (based on Off-Policy RL)**: Secondly, we propose a variant DePO-IW (introduced at the end of Sec. 4.2). In this variant, we further formalize the short-to-long context capability transfer as an off-policy reinforcement learning problem. Specifically, we treat the trajectories generated in the short-context environment as off-policy data. By leveraging importance sampling (Eq. 13), we then model the optimization of the long-context policy as an offline RL task. This provides an alternative perspective for achieving cross-context capability transfer.

---

> ### Author Response · Authors · 2025-11-22
> **Response to Reviewer oY4b (4/5)**
>
> > **Q1:** How long does the KD algorithm take to run in practice? Does this represent significant computational overhead compared to training the reinforcement learning methods?
>
> **A4:** Thank you for your insightful question. A key contribution of our DePO is that it not only enhances model performance but also reduces the overall computational overhead. To elucidate the computational advantages, we provide a comparative analysis of standard Long-context RL and our DePO:
>
> * **Standard Long-context RL**: The computational cost of this approach is dominated by two expensive steps:
>   1. Online Rollouts: This requires generating multiple reasoning trajectories within the long-context setting, a process we have shown to be highly time-consuming.
>   2. Long-context Optimization: This involves calculating gradients through all sampled reasoning trajectories within the long-context environment.
> * **Our DePO**: We decompose the computational cost into two distinct objectives:
>   1. Proxy Objective: All online sampling and the RL-based optimization are confined to a **short-context** setting, rendering this stage highly efficient.
>   2. Knowledge Distillation: This step completely **obviates the need for online sampling in the long context**. It only requires optimization through high-quality trajectories (generated in the short context and filtered via conditional computation) within the long-context environment.
>
> This accounts for the results presented in Fig. 5, where our approach demonstrates a 2.8% enhancement in performance relative to direct long-context reinforcement learning, coupled with a significant 49.5% decrease in per-step training duration.
> > **Q2:** In Table 3, neither DePO nor DePO-IW consistently outperform each other. Under which circumstances would a user want to use DePO-IW instead of DePO? Are there particular problem characteristics that might indicate whether one or the other would perform better?
>
> **A5:** Thank you for your insightful question. DePO and DePO-IW embody two distinct strategies for knowledge distillation:
>
> DePO: DePO employs the forward Kullback-Leibler (FKL) divergence to minimize the discrepancy between the model's output distributions under long and short contexts. This alignment is achieved through a conditional computation strategy, which ensures both stability and efficiency during training.
>
> DePO-IW (DePO with Importance-weighting): DePO-IW offers a more unified approach by situating the entire process within a RL framework. It reformulates the knowledge distillation step as an off-policy RL problem. Specifically, it treats trajectories generated from short-context as offline data and uses importance sampling to correct for the distribution shift (i.e., from the short-text proxy policy to the long-text target policy), thereby enabling policy updates on long texts.
>
> As shown in Table 3, when integrated with DAPO, DePO-IW$^\dagger$ achieves an average score of 55.0, surpassing the original DePO's 54.6. Similarly, with GSPO, DePO-IW$^\ddagger$ scores 54.3 on average, compared to DePO's 53.8. While the margin is modest, this consistent improvement across two different underlying RL algorithms suggests that DePO-IW is a more refined and potent variant. DePO-IW is particularly well-suited for problems where the output distribution is consistent between the short- and long-context versions of a task. This ensures that the policy distributions $\pi(\cdot|c_\text{short},x)$ and $\pi(\cdot|c_\text{long},x)$ are not excessively divergent. In this case, importance sampling is highly effective at bridging the gap. In contrast, the advantage of DePO lies in its computational efficiency. Instead of computing the importance sampling ratio for each trajectory, this method performs conditional computation, thereby reducing computational overhead. Consequently, its balance of effectiveness and efficiency makes DePO the recommended starting point for most applications.

---

> ### Author Response · Authors · 2025-11-22
> **Response to Reviewer oY4b (5/5)**
>
> > **Q3:** Lemma 3.1 does not seem to use any specifics of the long- or short-context settings, and appears to apply to any scenario in which we apply a policy to two different distributions. Does it apply only to the present long vs. short context setting, or can it be applied to these more general settings as well?
>
> **A6:** Thank you for your insightful question.
>
> DePO is a training framework tailored for long-context scenarios. Lemma 3.1 theoretically justifies the DePO objective as a principled method to bridge the cross-context performance gap. Notably, Lemma 3.1 carries an important implicit constraint: two input distributions share identical solution space. In our long-context setting, we have meticulously satisfied this prerequisite through careful data construction and by analyzing the invariance of reasoning patterns (as discussed before). However, we acknowledge that constructing such **proxy tasks** with identical solution spaces is highly challenging in more general settings.
>
> Nonetheless, we believe the underlying decomposed optimization paradigm of DePO holds significant potential for a broader range of long-context related tasks. For instance:
>
> * **Repository-level code completion:** The short-context could be function signatures and key dependencies, while the long-context is the entire repository.
> * **Object detection in computer vision:** The short-context could be a cropped image containing only the target object, while the long-context is the complex original scene containing that object.
>
> [1] RULER: What's the Real Context Size of Your Long-Context Language Models? COLM, 2024.
>
> [2] LongPO: Long context self-evolution of large language models through short-to-long preference optimization. ICLR, 2025.
>
> [3] SoLoPO: Unlocking Long-Context Capabilities in LLMs via Short-to-Long Preference Optimization. arXiv, 2025.
>
> ---
> We thank you again for your valuable feedback and sincerely hope this response addresses your concerns. We would be pleased to engage in any further discussion.

---

> ### Author Response · Authors · 2025-11-26
> **Gratitude and Awaiting Your Feedback**
>
> Dear Reviewer oY4b,
>
> Thank you once again for your time and for your valuable feedback on our manuscript.
>
> We have carefully considered all of your comments and have provided a comprehensive response. The corresponding revisions have been made in the manuscript, and we believe these changes have significantly improved our work.
>
> We sincerely hope that our responses and the revised manuscript will resolve your concerns. We are happy to engage in any further discussion.
>
> Best regards,
>
> Authors

---

### Author Response · Authors · 2025-11-26
**Response and Updated Manuscript to All Reviewers**

We would like to extend our sincerest gratitude to all reviewers for their thorough and insightful evaluation of our manuscript. We are greatly encouraged that the reviewers recognized several core strengths of our work, which we believe form its primary contributions. Specifically, we are pleased they highlighted:
- **Novelty and Conceptual Soundness:** The novelty and intuitive appeal of our DePO framework, which reframes a complex problem into a more tractable form (Reviewers `8Zdd`, `b3vd`, `szUZ`).
- **Comprehensive and Rigorous Experimentation:** The thoroughness and breadth of our empirical validation across ten datasets, multiple foundation models, and broad baselines (Reviewers `oY4b`, `b3vd`, `szUZ`).
- **Practical and Promising Framework:** This is underscored by its compatibility with diverse RL algorithms and its efficiency in reducing training overhead, making it a promising and scalable solution (Reviewers 8Zdd, b3vd).
- **High-Quality Presentation and Clarity:** The overall quality of the manuscript's writing, structure, and visual presentation, which makes the core ideas easy to follow (Reviewers `8Zdd`, `szUZ`).

In response to reviewers comments, we have undertaken revision of the manuscript. The key changes include:
- We provide a more detailed explanation of why and how reasoning capabilities learned from short contexts can be effectively generalized to long-context scenarios. (Reviewers `oY4b`, `szUZ`)
- We have meticulously clarified the technical details of our approach. This includes a more precise description of the knowledge distillation process, the joint training mechanism, and the mathematical formulation of our DePO and DePO-IW algorithms.  (Reviewers `oY4b`, `8Zdd`)
- We have re-rendered Fig 5. The new figure offers a clearer and more accurate visual representation of our findings. (Reviewer `8Zdd`)
- We have incorporated a new section dedicated to Future Work. This section explores serveral potential applications for the DePO and outlines promising research trajectories.  (Reviewers `oY4b`, `8Zdd`)
- We have included a description of the baseline data construction process.  (Reviewer `szUZ`)

We believe these revisions have substantially improved the clarity, depth, and impact of our paper. We thank you again for your valuable feedback and sincerely hope this response addresses your concerns. We would be pleased to engage in any further discussion.

---

### Meta-Review · Area_Chair_33nf · 2026-01-08

**Summary:**

This submission introduces Decomposed Policy Optimization (DePO), a framework that decouples long-context policy optimization into parallel short-context RL pattern acquisition and long-context knowledge distillation transfer. DePO addresses the key limitations of naive long-context RL, including high compute cost and degraded sampling quality of reasoning trajectories. Extensive results demonstrate that the framework shows strong generalization, algorithmic compatibility, and scalability, offering a practical path toward efficient long-context reasoning improvement.

**Reviewer Concerns:**

The reviewers' concerns can be summarized into the following parts:

1) Weak motivation for transfer premise: The intuition that reasoning learned on short contexts should transfer to long contexts is overly brief, despite being the core premise. A deeper, task-grounded argument for transfer effectiveness is missing, along with clearer articulation of the actual technical challenges being solved.

2) Limited technical novelty and variant confusion: The RL formulations largely restate prior work with minimal new insight, and multiple method variants (GROP, DAPO, GSPO, GSPO-IW, etc.) are not clearly disentangled across results. It is unclear which variant is used in each table/figure, how training time in Figure 5 is measured, and whether bars are plotted on a comparable scale.

3) Under-specified KD design and off-policy transfer risks: Key KD details are missing, including when and how often distillation is applied, whether standard KD algorithms suffice or require adaptation, and the practical runtime/overhead relative to RL baselines. The premise further assumes every long input has a clean short counterpart, which limits generalizability and creates unclear guidance for choosing DePO vs. DePO-IW.

4) Strategic and evaluation isolation gaps: The short-context solution is off-policy to long-context reasoning and may omit information present only in long inputs, risking reasoning mismatch and reward sparsity. The reward design and evaluator setups (including prompts, sampling strategy, scoring configs) are insufficiently isolated, and training interactions between “teacher” and “student” policies lack clarity, raising risks of collapse, bias drift, and credit-assignment bias.

**Reviewer Scores:**

The reviewers respectively rated the submission with the scores 4, 6, 4, 6, averaging 5 that is below the borderline score. Regarding the concerns, the authors provided substantial effort during the rebuttal to improve the submission. After reading the comments and the response, AC considered that after rebuttal, some reviewers may increase the scores given the addressed concerns like more explanations about the intuition, distillation process and the DePO-IW Algorithm, or more justification about the reward design. However, some concerns may remain, for example, 1) about Lemma 3.1, whether it is too general to specifically reflect the target, which is the core concern of the reviewer; 2) Two reviewers rating 4, both suffer from too many unclear details (although after rebuttal, it may be clear), which limited their further core questions about the judgement of the submission. For AC, it is hard to clearly take the acceptance decision towards this submission, given the competitive bar.

---

### Decision · Program_Chairs · 2026-01-26

Reject